# Short-term synaptic dynamics control the activity phase of neurons in an oscillatory network

**Diana Martinez[1†], Haroon Anwar[1], Amitabha Bose[2], Dirk M Bucher[1], Farzan Nadim[1,2]***

[1]Federated Department of Biological Sciences, New Jersey Institute of Technology and Rutgers University, Newark, United States; [2]Department of Mathematical Sciences, New Jersey Institute of Technology, Newark, United States

**Abstract** In oscillatory systems, neuronal activity phase is often independent of network frequency. Such phase maintenance requires adjustment of synaptic input with network frequency, a relationship that we explored using the crab, *Cancer borealis*, pyloric network. The burst phase of pyloric neurons is relatively constant despite a > two fold variation in network frequency. We used noise input to characterize how input shape influences burst delay of a pyloric neuron, and then used dynamic clamp to examine how burst phase depends on the period, amplitude, duration, and shape of rhythmic synaptic input. Phase constancy across a range of periods required a proportional increase of synaptic duration with period. However, phase maintenance was also promoted by an increase of amplitude and peak phase of synaptic input with period. Mathematical analysis shows how short-term synaptic plasticity can coordinately change amplitude and peak phase to maximize the range of periods over which phase constancy is achieved.
DOI: https://doi.org/10.7554/eLife.46911.001

**\*For correspondence:**
farzan@njit.edu

**Present address:** [†]Department of Biomedical Sciences and Dalton Cardiovascular Research Center, University of Missouri, Columbia, United States

**Competing interests:** The authors declare that no competing interests exist.

## Introduction

Oscillatory neural activity is often organized into different phases across groups of neurons, both in brain rhythms associated with cognitive tasks or behavioral states (*Hasselmo et al., 2002*; *Buzsáki and Wang, 2012*; *Buzsáki and Tingley, 2018*), and in central pattern generating (CPG) circuits that drive rhythmic motor behaviors (*Marder and Bucher, 2001*; *Marder et al., 2005*; *Grillner, 2006*; *Bucher et al., 2015*; *Katz, 2016*; *Stein, 2018*). The functional significance of different phases in the latter is readily apparent, as they for example provide alternating flexion and extension of limb joints, and coordination of movements between joints, limbs, and segments (*Krantz and Parks, 2012*; *Grillner and El Manira, 2015*; *Kiehn, 2016*; *Le Gal et al., 2017*; *Bidaye et al., 2018*). A hallmark of many such patterns is that the relative timing of firing between neurons is well maintained over a range of rhythm frequencies (*Dicaprio et al., 1997*; *Hooper, 1997b*; *Hooper, 1997a*; *Wenning et al., 2004*; *Marder et al., 2005*; *Grillner, 2006*; *Mullins et al., 2011*; *Le Gal et al., 2017*). If the latency of firing across different groups of neurons changes proportionally to the rhythm period, phase (latency over period) is invariant, in some cases providing optimal limb coordination at all speeds (*Zhang et al., 2014*).

The ability of the system to coordinate phases with changes in period arises from central coordinating mechanisms between circuit elements, as it is present in isolated nervous system preparations, but the underlying cellular and circuit mechanisms are not well understood. For instance, constant phase lags between neighboring segments in the control of swimming in lamprey fish and crayfish can be explained mathematically on the basis of asymmetrically weakly coupled oscillators, but the role of intrinsic and synaptic dynamics within each segment is unknown (*Cohen et al., 1992*;

*Skinner and Mulloney, 1998*; *Grillner, 2006*; *Mullins et al., 2011*; *Zhang et al., 2014*; *Le Gal et al., 2017*).

The pyloric circuit of the crustacean stomatogastric ganglion (STG) has inspired a series of experimental and theoretical studies of cellular and synaptic mechanisms underlying phase maintenance. The pyloric circuit generates a triphasic motor pattern with stable phase relationships over a wide range of periods (*Eisen and Marder, 1984*; *Hooper, 1997b*; *Hooper, 1997a*; *Bucher et al., 2005*; *Goaillard et al., 2009*; *Tang et al., 2012*; *Soofi et al., 2014*). Synapses in the pyloric circuit use graded as well as spike-mediated transmission (*Graubard et al., 1980*; *Harris-Warrick and Johnson, 2010*; *Zhao et al., 2011*; *Rosenbaum and Marder, 2018*). Follower neurons burst in rebound from inhibition from pacemaker neurons (*Marder and Bucher, 2007*; *Daur et al., 2016*), and post-inhibitory rebound delay scales with the period of hyperpolarizing currents (*Hooper, 1998*). Voltage-gated conductances slow enough for cumulative activation across cycles could promote such phase maintenance (*Hooper et al., 2009*). Similarly, short-term depression of graded inhibitory synapses is slow enough to accumulate over several pyloric cycles, meaning that effective synaptic strength increases with increasing cycle period (*Manor et al., 1997*; *Nadim and Manor, 2000*).

Theoretical studies have shown that short-term synaptic depression, by increasing inhibition strength with cycle period, should promote phase maintenance (*Manor et al., 2003*; *Mouser et al., 2008*), particularly in conjunction with inactivating (A-type) potassium currents (*Bose et al., 2004*; *Greenberg and Manor, 2005*), which control the rebound delay (*Harris-Warrick et al., 1995b*; *Harris-Warrick et al., 1995a*; *Kloppenburg et al., 1999*). These predictions remain experimentally untested.

Additionally, postsynaptic responses also depend on the actual trajectory of synaptic conductances, which are shaped by presynaptic voltage trajectories and short-term synaptic plasticity (*Manor et al., 1997*; *Mamiya et al., 2003*; *Zhao et al., 2011*; *Tseng et al., 2014*). If amplitude, duration, and trajectory of synaptic conductance determine rebound delay, phase maintenance necessitates all three of these parameters to change with cycle period in coordination. We used the dynamic clamp technique to exhaustively explore the range of these parameters and understand how the coordinated changes in synaptic dynamics determines the phase of follower neurons in an oscillatory circuit. Our findings are consistent with a mathematical framework that accounts for the dependence of amplitude and peak phase of the synaptic conductance on cycle period.

## Results

### Phase maintenance and latency maintenance

The firing of neurons in oscillatory networks is shaped by a periodic synaptic input. The relative firing latency of such neurons is often measured relative to a defined reference time in each cycle of oscillation, and is used to determine the activity phase of the neuron (see, for example *Belluscio et al., 2012*). For example, in a simple network consisting of a bursting oscillatory neuron driving a follower neuron (*Figure 1A1*), at a descriptive level, the latency ($\Delta t$) of the follower neuron activity relative to the onset of the oscillator's burst onset may depend on the oscillation cycle period ($P$). In response to a change in period (say, to $P_2$), the follower neuron may keep constant latency ($\Delta t_2 = \Delta t$), or constant phase, that is modify its latency proportionally to the change in period ($\Delta t_2 / P_2 = \Delta t/P$; *Figure 1A2*). However, in many oscillatory systems, for example the pyloric circuit (*Hooper, 1997b*; *Hooper, 1997a*), the relationship between $L$ and $P$ falls between these two extremes.

We demonstrated this point in the pyloric follower LP neuron using the following protocol. We voltage clamped one of the pacemaker PD neurons and drove this neuron with its own pre-recorded waveform, but applied at five different cycle periods (also denoted $P$). This protocol entrained the pacemaker group at this period, which forced the follower LP neuron to obey the same period (*Figure 1B*). We then measured the latency ($\Delta t$) of the LP burst onset with respect to onset of the PD neuron burst. A plot of the LP latency $\Delta t$ or phase ($\Delta t/P$) for different cycle periods demonstrates the above-mentioned finding that the LP neuron activity falls between the two limits of constant phase and constant latency (*Figure 1C*).

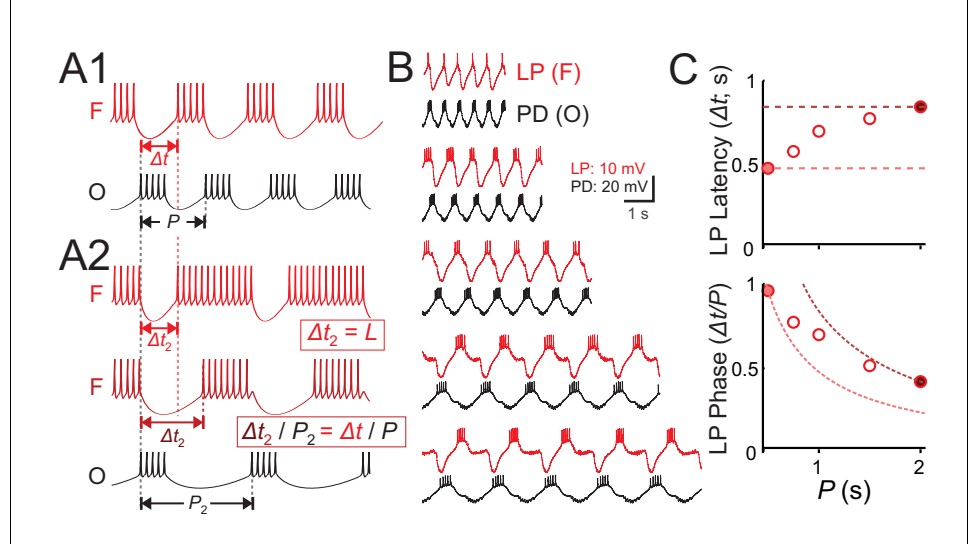

**Figure 1.** Latency constancy and phase constancy as a function of period. (**A1**) Schematic diagram showing that a follower neuron (F) strongly inhibited by a bursting oscillatory neuron (O) with period $P$ can produce rebound bursts with the same period at a latency $\Delta t$. (**A2**) If the period of O changes to a new value ($P_2$), the new F burst latency ($\Delta t_2$) typically falls between two extremes: it could stay constant (top trace) or change proportionally to $P_2$, so that the burst phase ($\Delta t/P$) remains constant (middle trace). (**B**) Example traces of the pyloric pacemaker PD neuron and the follower LP neuron represent the O and F relationship in panel A. Here, the PD neuron is voltage clamped and a pre-recorded waveform of the same neuron is used to drive this neuron to follow different cycle periods. The LP neuron follows the same period because of the synaptic input it receives. (**C**) A measurement of the LP neuron burst onset time ($\Delta t$) with respect to the onset of the PD neuron burst shows that $\Delta t$ falls between the two limits of constant latency and constant phase. Dotted curves represent constant latency matched to the latencies at the two extreme $P$ values.

DOI: https://doi.org/10.7554/eLife.46911.002

## The burst onset time of the LP neuron depends on the temporal dynamics of its input

The LP neuron does not have intrinsic oscillatory properties, but oscillates due to the synaptic input it receives from the pacemaker anterior burster (AB) and pyloric dilator (PD) neurons, and the follower pyloric constrictor (PY) neurons (***Figure 2A***). The burst onset phase of the LP neuron ($\varphi_{LP} = \Delta t/P$; ***Figure 2A***) is shaped by the interaction between synaptic inputs and the neuron's intrinsic dynamics that influence post-inhibitory rebound. We measured an overall burst onset phase of the LP neuron to be $\varphi_{LP} = 0.34 \pm 0.03$ (N = 9).

As a first-order quantification, we measured how inputs to the LP neuron interact with its intrinsic properties to determine the timing between its bursts, in the absence of network oscillations. To this end, we blocked the synaptic input from the pacemaker AB and follower PY neurons to the LP neuron (***Figure 2B***) and drove the LP neuron with a noise current input (see Materials and methods). In response to the noise input, the LP neuron produced an irregular pattern of spike times, which included a variety of bursting patterns with different spike numbers (***Figure 2C***). We were interested in the characteristics of inputs producing different burst onset latencies. However, unlike a periodic input, noise input does not provide a well-defined reference point to measure the burst onset latency. We categorized bursts with respect to the preceding inter-burst intervals (IBIs; see Materials and methods), during which no other action potentials occurred. We classified these IBIs in bins (300, 500, 700 and 900 ms) and tagged bursts based on the IBI values (***Figure 2C***). We characterized the driving input leading to bursts with specific IBIs by burst-triggered averaging the input current ($I_{BTA}$; an example shown in ***Figure 2D***). Our analysis produced a single $I_{BTA}$ for each of the four IBIs in each preparation (N = 23). $I_{BTA}$'s of each preparation were first normalized in amplitude by the (negative) peak value of the $I_{BTA}$ at IBI = 300 ms (***Figure 2E***; average shown in ***Figure 2F***) to examine how peak amplitude ($I_{peak}$) varied with IBI. These data were then normalized in time

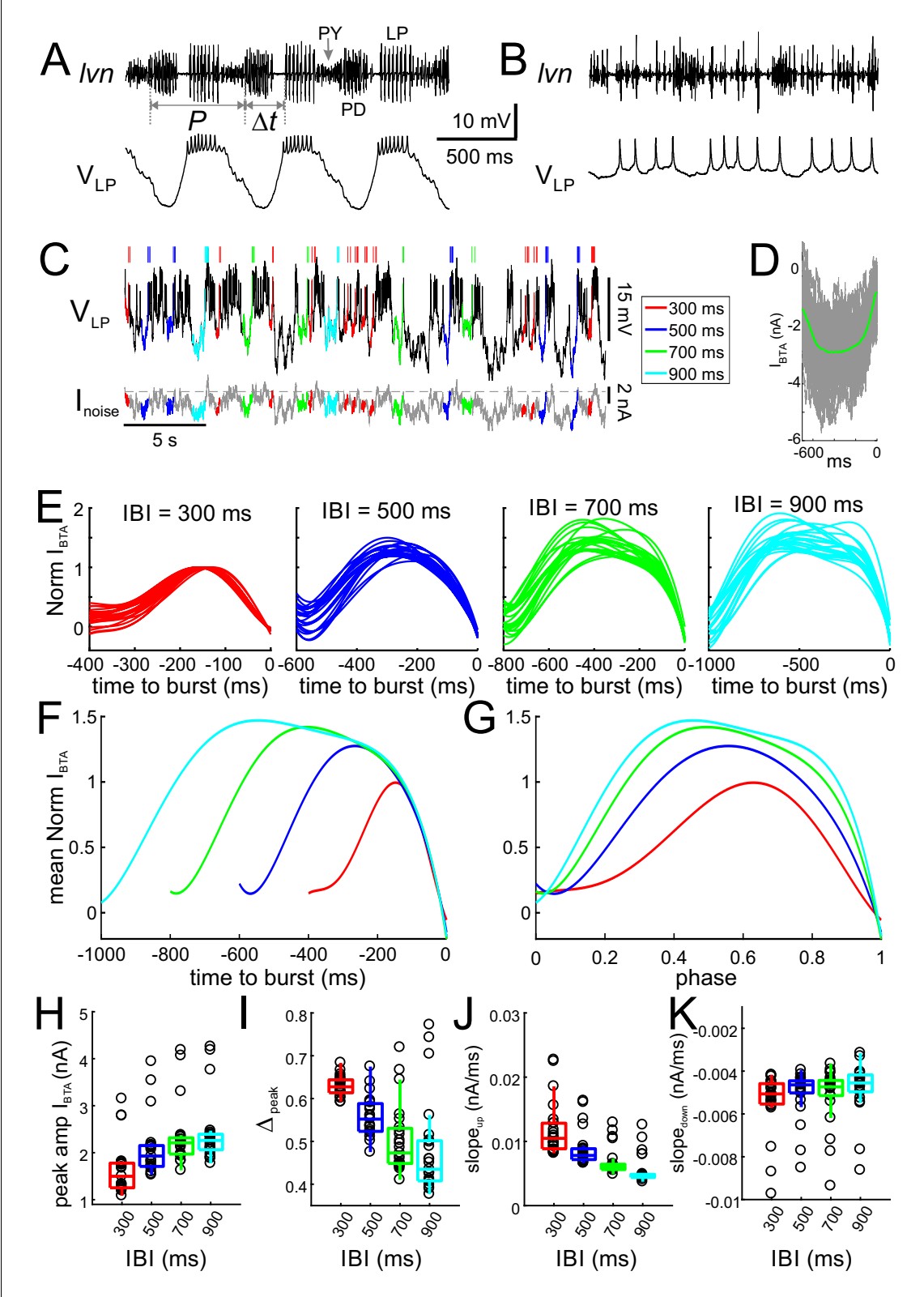

**Figure 2.** Inputs to the LP neuron influence burst time, spike number and interval. (**A**) Simultaneous intracellular recording of the LP neuron and extracellular recording of the lateral ventricular nerve (*lvn*), containing the axons of the LP, PD and PY neurons (arrows). Period (*P*) and the burst onset time (*Δt*) of the LP neuron are defined in reference to the pacemaker group (PD) burst. (**B**) Blocking the AB and PY synaptic inputs (10 μM picrotoxin) to the LP neuron disrupts its bursting oscillations. (**C**) The LP neuron, in picrotoxin, was driven with a noise current input ($I_{noise}$) for 60 min. In response, the
*Figure 2 continued on next page*

*Figure 2 continued*

LP neuron produced an irregular pattern of bursting. Specific inter-burst intervals (*IBI*s) were tagged and used for burst-triggered averaging. (D) Example of burst-trigger-averaged input current (*I_BTA*, green). Individual traces are shown in gray. (E) For each *IBI* (300, 500, 700, 900 ms), *I_BTA* was calculated and normalized to the (negative) peak value of *I_BTA* for IBI = 300 ms. Different traces in each panel show the *I_BTA* of different preparations. (F) The mean (across preparations) of the normalized *I_BTA*s shown in panel E. (G) Traces in panel F normalized by *IBI*. (H–K) Four parameters define the shape of the *I_BTA*: peak amplitude $I_{amp}$ (H), peak phase $\Delta_{peak}$ (I), slope$_{up}$ (J) and slope$_{down}$ (K) across preparations. IBI had a significant effect on amplitude $I_{amp}$ (p<0.001), peak phase $\Delta_{peak}$ (p<0.001), slope$_{up}$ (p<0.001) and slope$_{down}$ (p=0.002).

DOI: https://doi.org/10.7554/eLife.46911.003

The following source data is available for figure 2:

**Source data 1.** This Excel file contains four sheets, including all measured attributes of the burst-triggered average current (I_BTA) for different IBIs (N = 23) as shown in *Figure 2H–2K*.

DOI: https://doi.org/10.7554/eLife.46911.004

(*Figure 2G*) to examine the effect of IBI on peak phase ($\Delta_{peak}$) and the rise (*slope_{up}*) and fall (*slope_{down}*) slopes of the input current across preparations. We found that IBI had a significant effect on $I_{peak}$, $\Delta_{peak}$, *slope_{up}* and *slope_{down}* (all one-way RM-ANOVA on ranks; data included in *Figure 2— source data 1*). In particular, larger IBIs corresponded to larger $I_{peak}$ values (*Figure 2F–2H*; p<0.001, $\chi^2 = 65.87$) with smaller (more advanced) $\Delta_{peak}$ (*Figure 2I*; p<0.001, $\chi^2 = 41.35$). The change in $\Delta_{peak}$ was due to a decrease in *slope_{up}* (p<0.001, $\chi^2 = 65.25$), whereas *slope_{down}* did not vary as much (*Figure 2J–2K*; p=0.002, $\chi^2 = 14.77$).

## The burst onset phase of the LP neuron oscillation depends on its synaptic input

Injection of noise current revealed that the timing of the LP response is exquisitely sensitive to the duration and amplitude of inputs. In the intact system, the primary determinant of input duration and amplitude is the network period, as increasing period increases both presynaptic pacemaker burst duration (*Hooper, 1997b*; *Hooper, 1997a*) and synaptic strength (*Manor et al., 1997*; *Nadim and Manor, 2000*). To explore the effect of the duration and strength of the synaptic input, we used dynamic clamp to drive the LP neuron with a realistic synaptic conductance waveform.

We constructed this realistic waveform by measuring the synaptic current input to the LP neuron during ongoing pyloric oscillations (*Figure 3A*). These measurements showed the two components of inhibitory synaptic input: those from the pacemaker AB and PD neurons (left arrow) and those from the follower PY neurons (right arrow). In each cycle, the synaptic current always had a single peak, but the amplitude and phase of this peak showed variability across preparations (*Figure 3B*, average in blue).

The realistic conductance input was injected periodically with strength $g_{max}$ (*Figure 3C*). For any fixed $g_{max}$, $\varphi_{LP}$ decreased as a function of P (*Figure 3D*), that is the relative onset of the LP burst was advanced in slower rhythms. In contrast to the effect of P, for any given P, $\varphi_{LP}$ increased sublinearly as a function of $g_{max}$ (*Figure 3E*). *Figure 3F* combines the simultaneous influence of both parameters on $\varphi_{LP}$. The results shown in *Figure 3D* indicate that the LP neuron intrinsic properties alone do not produce phase constancy. However, level sets of $\varphi_{LP}$ (highlighted for three values in *Figure 3F*), indicate that phase could be maintained over a range of P values, if $g_{max}$ increases as a function of P. This finding was predicted by our previous modeling work, in which we suggested that short-term synaptic depression promotes phase constancy by increasing synaptic strength as a function of P (*Manor et al., 2003*; *Bose et al., 2004*). We will further discuss the role of synaptic depression below.

To clarify the results of *Figure 3*, it is worth examining the extent of phase maintenance for fixed $g_{max}$. An example of this is shown in *Figure 4A* (turquoise plots). A comparison of these data with the theoretical cases in which either delay or phase is constant suggests that the LP neuron produces relatively good phase maintenance, at least much better in comparison with constant delay. However, this conclusion is misleading because, in these experiments, the duty cycle of the synaptic input was kept constant. Therefore, most of the phase maintenance is due the fact that the synaptic input keeps perfect phase. In fact, if the reference point measures phase relative to the end –rather than onset– of the PD burst (*Figure 4B*), phase maintenance of the LP neuron is barely better than in the constant delay case (*Figure 4A*, purple plots). It is therefore clear that phase maintenance by the LP

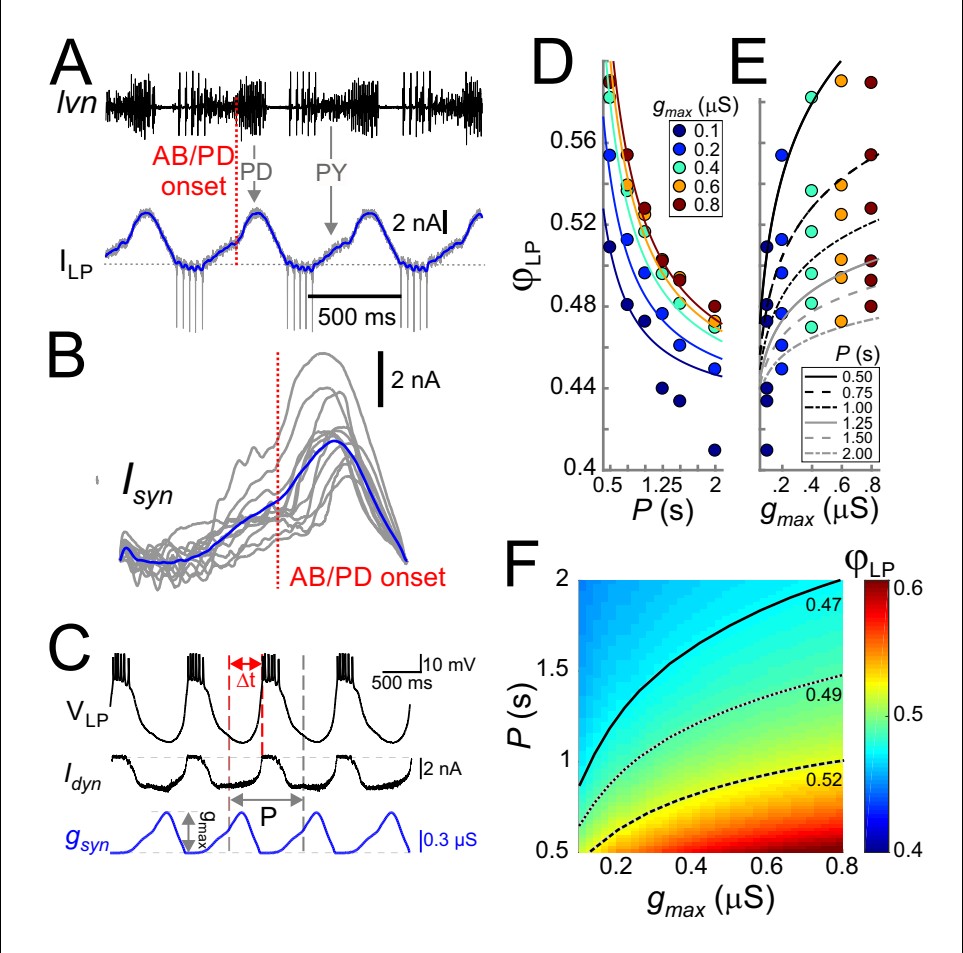

**Figure 3.** Cycle period and synaptic strength affect the phase of LP burst onset in opposite directions. (A) The synaptic input to the LP neuron was measured by voltage clamping it at a holding potential of −50 mV during ongoing oscillations. The onset of the pacemaker (AB/PD) activity is seen as a kink in the synaptic current ($I_{LP}$, blue). Dashed line: 0 nA. (B) Synaptic input averaged across (last 5 of 30) cycles from nine different LP neurons. Traces are aligned to the onset of the PD neuron burst (dotted vertical red line; see panel A), normalized by the cycle period and terminated at the end of the downslope (coincident with the first LP action potential when present). The blue trace shows the average. (C) An example of the LP neuron driven by the realistic synaptic waveform in dynamic clamp. The burst onset time ($\Delta t$) was measured relative to the AB/PD onset and used to measure the LP phase ($\varphi_{LP}$). $g_{max}$ denotes the conductance amplitude. (D) Mean $\varphi_{LP}$ (N = 9 preparations) shown as a function of P and fit with the function given by **Equation (8)** (fit values $\tau_s$=26.0 ms, $g^*$=0.021 μS and $\Delta_{peak}\cdot DC$ = 0.43). (E) Mean $\varphi_{LP}$ plotted against $g_{max}$ also shown with the fit to **Equation (8)**. (F) Heat map, obtained from fitting **Equation (8)** to the data in panels D and E, shows $\varphi_{LP}$ as a function of both $g_{max}$ and P. Black curves show the level sets of phase constancy for three values of $\varphi_{LP}$ (0.47, 0.49, and 0.52).
DOI: https://doi.org/10.7554/eLife.46911.005

neuron would require the properties of the synaptic input to change as a function of P, a hallmark of short-term synaptic plasticity (*Fortune and Rose, 2001*; *Grande and Spain, 2005*). As mentioned above, short-term plasticity such as depression could produce changes in $g_{max}$ as a function of P. Independently of $g_{max}$, the peak time of the synaptic current is another parameter that could change with P and influence the timing of the postsynaptic burst. We therefore proceeded to systematically explore the influence of P, $g_{max}$ and the synaptic peak time on $\varphi_{LP}$.

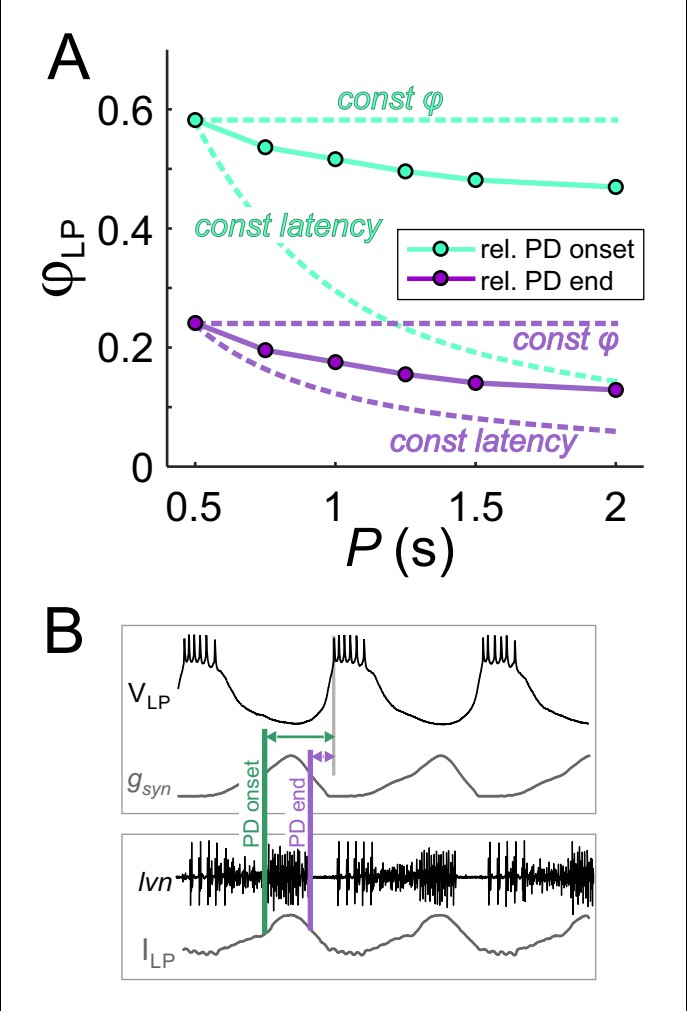

**Figure 4.** The constant duty cycle of synaptic conductance is a major factor in phase maintenance. (**A**) The change in $\varphi_{LP}$ values with $P$ are compared with the constant phase (solid curve) and constant latency (dashed curve) extremes. Lime traces show the usual values of $\varphi_{LP}$, calculated from the LP burst onset latency with respect to the onset of the PD burst. Lavender traces show $\varphi_{LP}$ calculated from the LP burst onset latency with respect to the end of the PD burst. Data shown are the same as in *Figure 3D* for $g_{max}$ = 0.4 µS. (**B**) Schematic diagram shows the latency of LP burst onset measured with respect to the (estimated) onset and end of the PD burst in the dynamic clamp experiments (see Materials and methods). Bottom panel shows the synaptic current waveform measured in the voltage-clamped LP neuron during ongoing pyloric activity. Top panel shows the dynamic clamp injection of the synaptic conductance waveform into the LP neuron. The current waveform of the bottom panel is aligned to the conductance waveform of the top panel for the comparison used in determining the PD burst onset and end in the top panel.

DOI: https://doi.org/10.7554/eLife.46911.006

## A systematic exploration of synaptic input parameters on the phase of the LP neuron

For a detailed exploration of the influence of the synaptic input on $\varphi_{LP}$, we approximated the trajectory of the (unitary) synaptic conductance in one cycle by a simple triangle (*Figure 5A*), which could be defined by three parameters: duration ($T_{act}$), peak time ($t_{peak}$) and amplitude ($g_{max}$) (*Figure 5B*). This simplified triangular synaptic conductance waveform could then be repeated with any period ($P$) to mimic the realistic synaptic input to the LP neuron. For a given synaptic duration $T_{act}$, the peak phase of the synapse can be defined as $\Delta_{peak} = t_{peak}/T_{act}$). The parameter $\Delta_{peak}$ is known to vary as a function of $P$ (*Tseng et al., 2014*) and, in a previous study, we found that $\Delta_{peak}$ may influence the

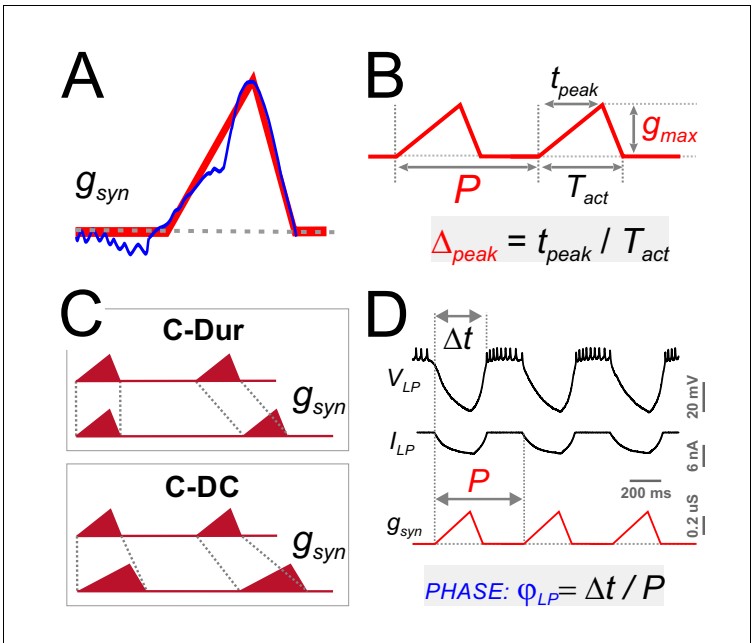

**Figure 5.** Four parameters describing synaptic shape were varied in the experimental paradigm. (**A**) A triangle-shaped conductance was used to mimic the synaptic input to the LP neuron. (**B**) The triangular waveform can be described by period (P), duration ($T_{act}$), peak time ($t_{peak}$) and amplitude ($g_{max}$). (**C**) In dynamic clamp runs, the synapse duration $T_{act}$ was kept constant at 300 ms (C-Dur) or maintained at a constant duty cycle ($T_{act}/P$) of 0.3 (C–DC) across all values of P. (**D**) Intracellular voltage recording of the LP neuron during a dynamic clamp stimulation run using the triangle conductance (in picrotoxin). The burst onset time (Δt, calculated in reference to the synaptic conductance onset) was used to calculate the activity phase ($\varphi_{LP} = \Delta t/P$).

DOI: https://doi.org/10.7554/eLife.46911.007

activity of the postsynaptic neuron, independent of P and $g_{max}$ (*Mamiya and Nadim, 2004*). We therefore systematically explored the influence of three parameters of the synaptic input (P, $g_{max}$ and $\Delta_{peak}$) on $\varphi_{LP}$.

As with the realistic synaptic waveforms (*Figure 3*), we used the dynamic clamp technique to apply the triangular conductance waveform periodically to the LP neuron in the presence of the synaptic blocker picrotoxin. Across different runs within the same experiment, the parameters P, $g_{max}$ and $\Delta_{peak}$ were changed on a grid (see Materials and methods). In addition, all combinations of these three parameter values were run in two conditions in the same experiment, 1: with constant duration, that is constant $T_{act}$ across different P values (C-Dur of 300 ms), and 2: with constant duty cycle, that is $T_{act}$ changing proportionally to P (C-DC of 0.3; *Figure 5C*). Using these protocols, we measured the effects of synaptic parameters on $\varphi_{LP}$ (*Figure 5D*).

The LP neuron produced burst responses that followed the synaptic input in a 1:1 manner across all values of P that were used (*Figure 6A1*). When $g_{max}$ and $\Delta_{peak}$ were kept constant, $\varphi_{LP}$ decreased as a function of P (*Figure 6A2*). This decrease was always larger for the C-Dur case than the C-DC case. For both C-DC and C-Dur, this trend was seen across all values of $\Delta_{peak}$ and $g_{max}$ (*Figure 6A3*). The effect of P on $\varphi_{LP}$ was highly significant for both C-DC (three-way ANOVA, p<0.001, F = 100.677) and C-Dur (three-way ANOVA, p<0.001, F = 466.424), indicating that the period and duration of the inhibitory input to the LP neuron had a significant effect on its phase.

Changing $g_{max}$ produced a large effect on the level of hyperpolarization in the LP neuron, but this usually translated to only a small or modest effect on the time to the first spike following inhibition (*Figure 6B1*). Overall, increasing $g_{max}$ at constant values of P and $\Delta_{peak}$ produced a significant but only small to moderate increase in $\varphi_{LP}$ (three-way ANOVA, p<0.001, F = 10.798). Although increasing $g_{max}$ produced the same qualitative effect for both the C-DC and C-Dur (e.g., *Figure 6B2*), $\varphi_{LP}$ in the C-DC case was restricted to a smaller range (*Figure 6B3* top vs. bottom panels). Overall, this increase was robust for most values of P and $\Delta_{peak}$ (*Figure 6B3*).

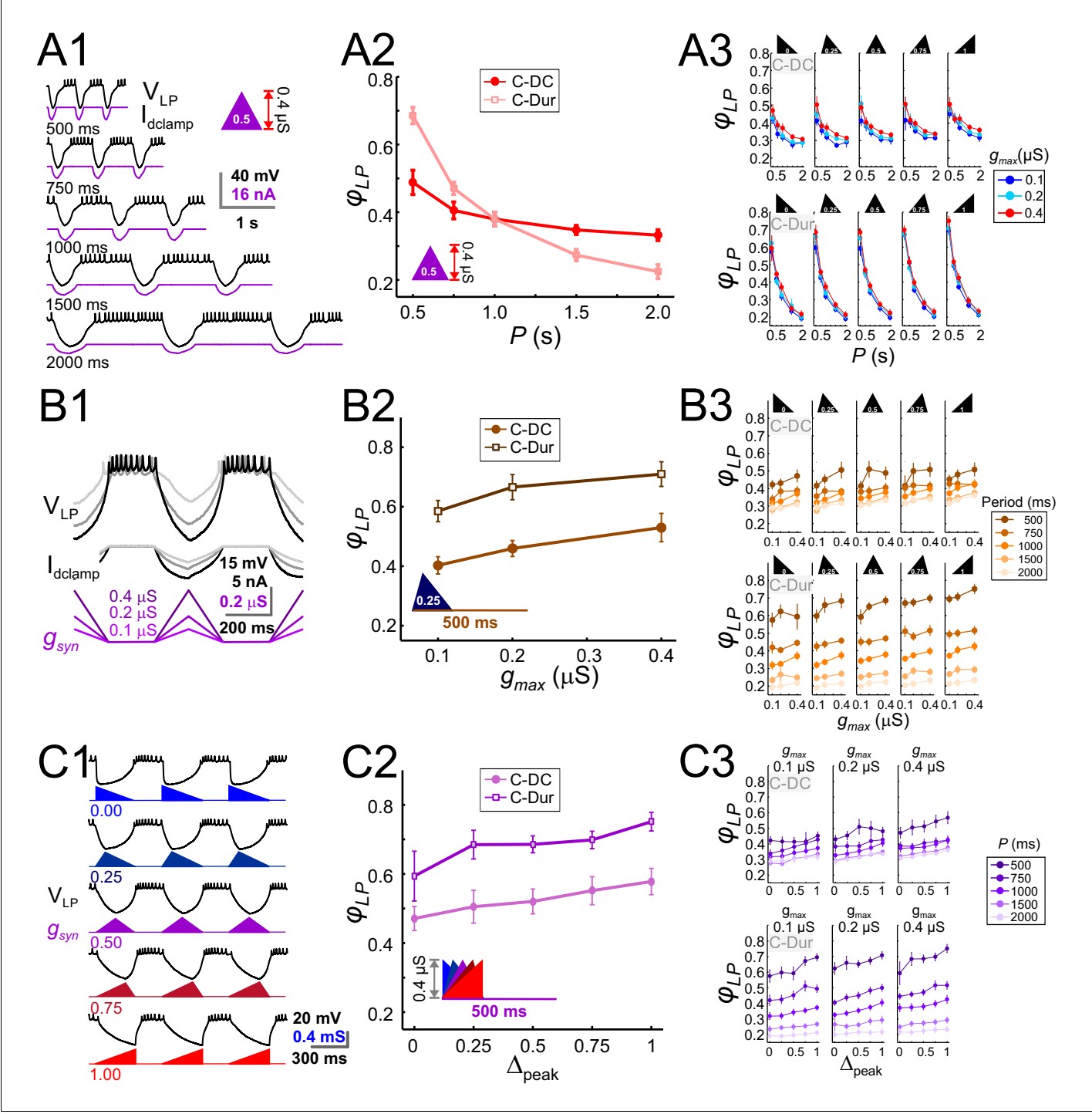

**Figure 6.** The LP burst onset phase decreases as a function of P, but increases as a function of $g_{max}$ and $\Delta_{peak}$. Periodic injection of an inhibitory triangular waveform conductance into the LP neuron (in picrotoxin) produced bursting activity from which $\varphi_{LP}$ was calculated. The parameters $g_{max}$, $\Delta_{peak}$ and P were varied across runs for both C-Dur and C-DC cases. (**A**) $\varphi_{LP}$ decreases as a function of P. (**A1**) Intracellular recording of an LP neuron showing a C-DC conductance input across five periods. (**A2**) $\varphi_{LP}$ for the example shown in A1 plotted as a function of P (for $g_{max}$ = 0.4 μS, $\Delta_{peak}$=0.5) for both C-Dur and C-DC cases. $\varphi_{LP}$ decreases rapidly with P and the drop is larger for the C-Dur case. (**A3**) $\varphi_{LP}$ decreased with P in both the C-DC case (three-way RM ANOVA, p<0.001, F = 100.7) and the C-Dur case (three-way RM ANOVA, p<0.001, F = 466.4) for all values of $\Delta_{peak}$. The range of $\varphi_{LP}$ drop was greater for the C-Dur case compared to the C-DC case. (**B**) $\varphi_{LP}$ increases as a function of $g_{max}$. (**B1**) Intracellular recording of an LP neuron showing the conductance input across three values of $g_{max}$. (**B2**) $\varphi_{LP}$ for the example shown in B1 plotted as a function of P (for p=500 ms, $\Delta_{peak}$=0.25) shows a small increase for both C-Dur and C-DC cases. (**B3**) $\varphi_{LP}$ increased with $g_{max}$ in almost all trials for both C-DC and C-Dur cases and all values of

*Figure 6 continued on next page*

*Figure 6 continued*
$\Delta_{peak}$. (C) $\varphi_{LP}$ increases as a function of $\Delta_{peak}$. (C1) Intracellular recording of the LP neuron showing the conductance input for five values of $\Delta_{peak}$. (C2) $\varphi_{LP}$ for the example neuron in C1 plotted as a function of $\Delta_{peak}$ (for p=500 ms, $g_{max}$ = 0.4 μS) for both C-DC and C-Dur cases. (C3) $\varphi_{LP}$ increased with $\Delta_{peak}$ for both C-DC and C-Dur cases and for all values of $g_{max}$. In all panels, error bars show standard deviation.
DOI: https://doi.org/10.7554/eLife.46911.008

Increasing $\Delta_{peak}$ for a constant value of $P$ and $g_{max}$ (*Figure 6C1*), produced a small but significant increase in $\varphi_{LP}$ (three-way ANOVA, p<0.001, F = 17.172). This effect was robust for most values of $P$ and $g_{max}$, for both C-DC and C-Dur (*Figure 6C2 and C3*).

These results showed that all three parameters that define the shape of the IPSC influence $\varphi_{LP}$. Clearly, the strongest effect is the decrease in $\varphi_{LP}$ as a function of $P$. However, $\varphi_{LP}$ modestly increases as a function of the other two parameters, $g_{max}$ and $\Delta_{peak}$. This raised the question how $g_{max}$ and $\Delta_{peak}$ would have to change in coordination as a function of $P$ to counteract the effect of $P$ on $\varphi_{LP}$ and achieve phase constancy.

## Coordinated changes of $g_{max}$ and $\Delta_{peak}$ produce the largest effect on phase

To explore how $g_{max}$ and $\Delta_{peak}$ might interact to influence $\varphi_{LP}$, we examined the sensitivity of $\varphi_{LP}$ to these two parameters, individually and in combination, for all values of $P$ in our data (see Materials and methods). Sensitivity of $\varphi_{LP}$ to these two parameters varied across $P$ values, with larger sensitivity at lower values of $P$ (two-way RM-ANOVA, p<0.001, F = 16.054; data included in *Figure 7—source data 1*). For simplicity, we averaged the sensitivity values across different $P$ values to obtain an overall measure of the influence of $g_{max}$ and $\Delta_{peak}$. These results showed that, for the C-DC case, $\varphi_{LP}$ had a positive sensitivity to $g_{max}$ and a smaller positive sensitivity to $\Delta_{peak}$ (*Figure 7A*). The sensitivity was largest if the two parameters were varied together ($g_{max}$ + $\Delta_{peak}$) and smallest if they were varied in opposite directions ($g_{max}$ - $\Delta_{peak}$; two-way RM-ANOVA, p<0.001, F = 3.330). Similarly, these sensitivity values were also significantly different for the C-Dur case (*Figure 7B*; two-way RM-ANOVA, p<0.001, F = 2.892), with largest sensitivity for $g_{max}$ + $\Delta_{peak}$ and smallest for $g_{max}$ - $\Delta_{peak}$.

## Level sets of $\varphi_{LP}$ in the P-$g_{max}$-$\Delta_{peak}$ space for C-DC and C-Dur cases

To search for phase constancy across different $P$ values in our dataset, we expressed $\varphi_{LP}$ as a function of the three IPSC parameters, $P$, $g_{max}$ and $\Delta_{peak}$: $\varphi_{LP} = \Phi(P, g_{max}, \Delta_{peak})$. *Figure 8* shows heat map plots of the function $\Phi$, plotted for the range of values of $P$ and $\Delta_{peak}$ and four values of $g_{max}$. In these plots, phase constancy can be seen as the set of values in each graph that are isochromatic, indicating the level sets of the function $\Phi$. These level sets are mathematically defined as hypersurfaces on which the function has a constant value: $\Phi(P, g_{max}, \Delta_{peak}) = \varphi_c$. For the C-DC case, in each $g_{max}$ section of the plot, the level sets (e.g. $\varphi_c$=0.34 denoted in white) spanned a moderate range of $P$ values as $\Delta_{peak}$ increased (*Figure 8A1*). The span of $P$ values across all four panels indicates the range of cycle periods for which phase constancy could be achieved by varying $g_{max}$ and $\Delta_{peak}$. This range of $P$ values (spanned by the white curves) was considerably smaller for the C-Dur case (*Figure 8A2*).

For any constant phase value $\varphi_c$, these level sets can be expressed as

$$P = P_{\varphi_c}(g_{max}, \Delta_{peak}),$$

which describes a surface in the 3D space, yielding the $P$ value for which phase can be maintained at $\varphi_c$, for the given values of $g_{max}$ and $\Delta_{peak}$. The level set indicated by the white curves in panel A for the C-DC case is plotted as a heat map in *Figure 8B1* and can be compared with the same plot for the C-Dur case in *Figure 8B2*. The range of colors in each plot (marked next to each panel) indicates the range of $P$ values for which phase can be kept at $\varphi_c$=0.34. To reveal how this range depends on the desired phase, we measured this range for all values of $\varphi_c$ between 0.2 and 0.8 (*Figure 8C1 and C2*). We found that the LP neuron could not achieve phases below 0.3 in the C-DC case (*Figure 8C1*), which is simply because the neuron never fired during the inhibitory synaptic current (which had a duty cycle of 0.3). Furthermore, the range of $P$ values for which the LP phase could be

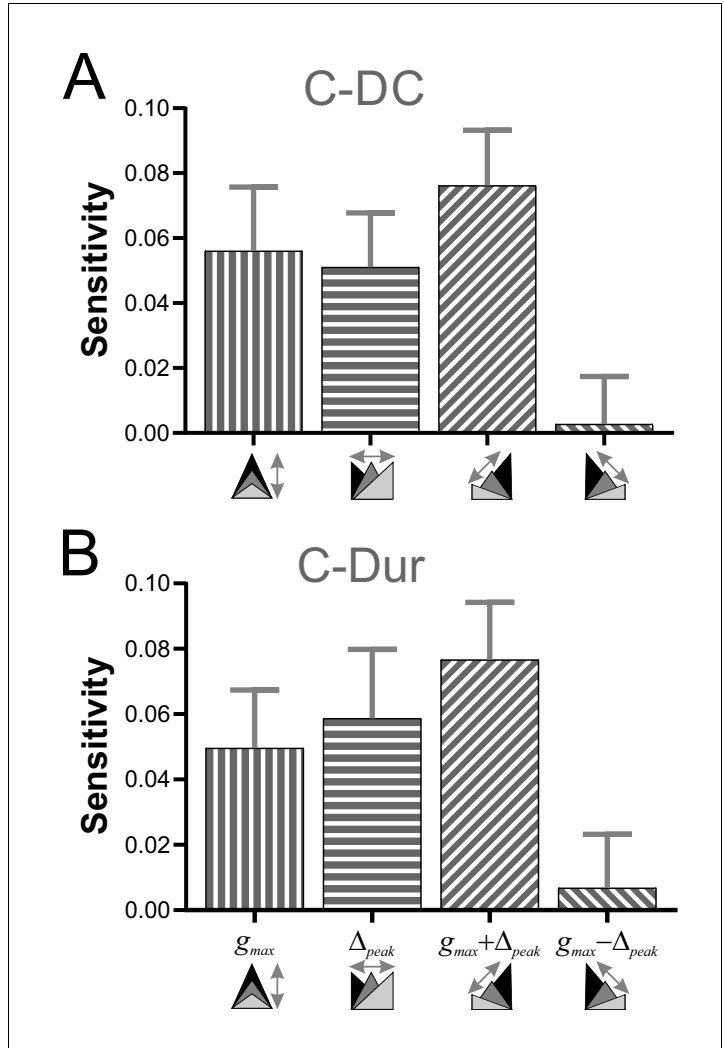

**Figure 7.** Sensitivity analysis shows that $\varphi_{LP}$ increases more effectively if $g_{max}$ and $\Delta_{peak}$ increase together. (**A**) The sensitivity of $\varphi_{LP}$ to local changes in $g_{max}$ and $\Delta_{peak}$ was averaged across all values of $P$ for the C-DC case. Sensitivity was largest if both parameters were increased together ($g_{max} + \Delta_{peak}$) and smallest if they were varied in opposite directions ($g_{max} - \Delta_{peak}$; one-way RM-ANOVA, p<0.001, F = 3.330). (**B**) The same sensitivity analysis in the C-Dur case shows similar results (one-way RM-ANOVA, p<0.001, F = 2.892). In both panels, error bars show standard deviation.

DOI: https://doi.org/10.7554/eLife.46911.009

The following source data is available for figure 7:

**Source data 1.** This Excel file contains two sheets for the C-DC and C-Dur cases.
DOI: https://doi.org/10.7554/eLife.46911.010

maintained by varying $g_{max}$ and $\Delta_{peak}$ was much larger for C-DC inputs compared to C-Dur Inputs, for all $\varphi_c$ values between 0.31 and 0.54.

## A model of synaptic dynamics could predict activity onset phase of the LP neuron

To gain a better understanding of our experimental results, we derived a mathematical description of the phase of a follower neuron such as LP, based on the following assumptions: 1, that the firing time of this neuron was completely determined by its synaptic input, 2, that in each cycle the synaptic conductance $g_{syn}$ increased to a maximum value $g_{max}$ for a time interval $T_{act}$ (the active duration of the synapse) and decayed to 0 otherwise, and 3, that the follower neuron remained inactive when

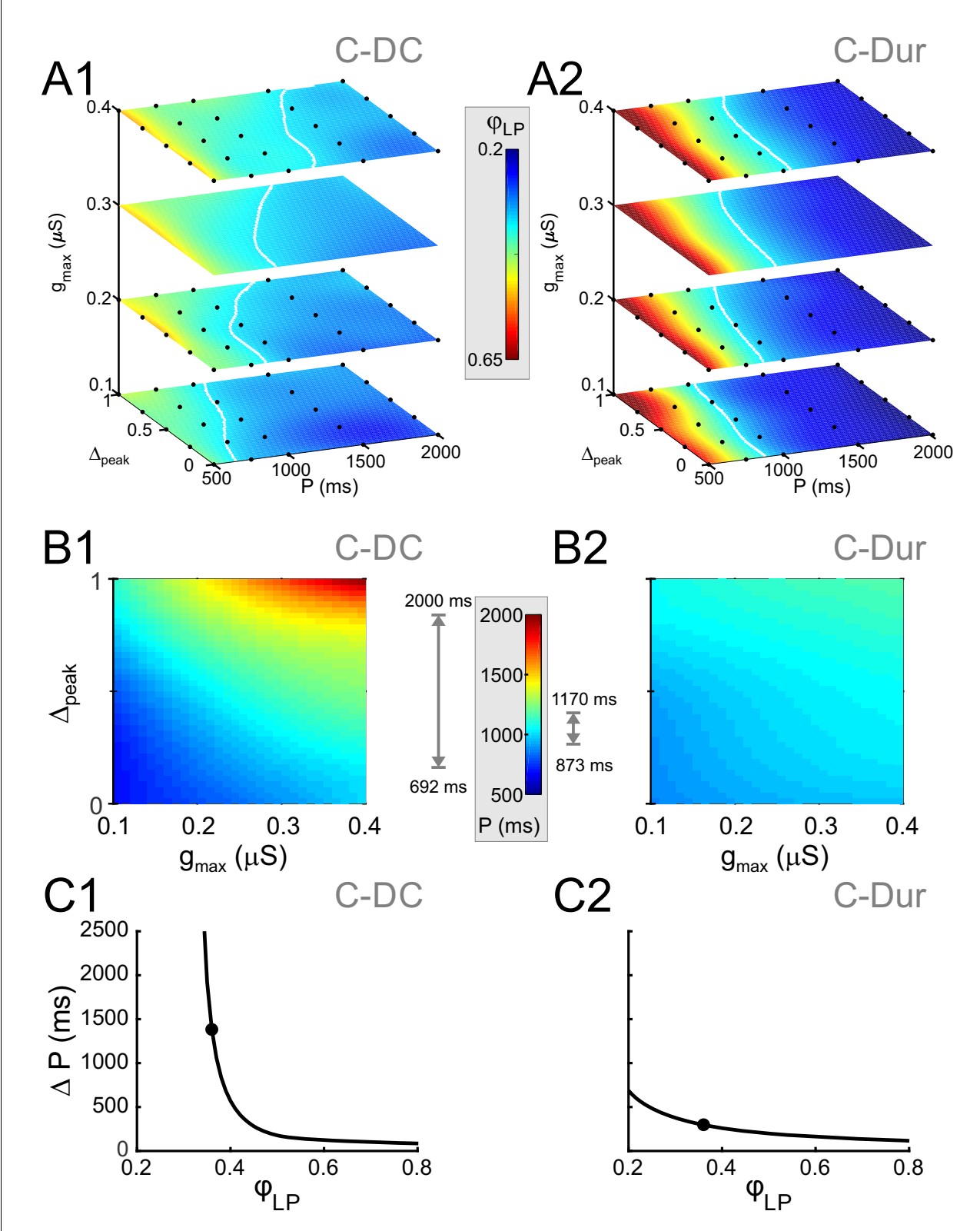

**Figure 8.** Simultaneous increase of both $\Delta_{peak}$ and $g_{max}$ across their range of values can produce phase maintenance across a large P range in the C-DC case and a much smaller P range in the C-Dur case. (**A**) Heat map plots of the function $\Phi$ (see Materials and methods), plotted for the range of values of $P$ and $\Delta_{peak}$ and 4 values of $g_{max}$ for the C-DC (**A1**) and C-Dur (**A2**) cases. The white curves show the level set of $\varphi_{LP}$=0.34, shown as an example of phase constancy. The color maps are interpolated from sampled data (see Materials and methods; N = 9 experiments). The locations of the

*Figure 8 continued on next page*

*Figure 8 continued*

sampled data are marked by black dots. (B) Heat map for the level sets $\varphi_{LP}$=0.34 for the C-DC (B1) and C-Dur (B2) cases. Range of colors in each panel indicate the range of $P$ values for which $\varphi_{LP}$ could remain constant at 0.34 for each case, as indicated by the gray arrows on the side of the heatmap color legend. (C) The range ($\Delta P$) of $P$ values for which $\varphi_{LP}$ could remain constant at any value between 0.2 and 0.8 for the C-DC (C1) and C-Dur cases (C2). Filled circles show the values shown in panel B. The LP neuron cannot achieve $\varphi_{LP}$ values below 0.3 in the C-DC case. For $\varphi_{LP}$ values between 0.3 and ~0.65, the range was larger in C-DC case.

DOI: https://doi.org/10.7554/eLife.46911.011

$g_{syn}$ was above some threshold $g^*$. The derivation of this model is described in the Materials and methods.

This simple model provided a mathematical description of $\varphi_{LP}$ as a function of $P$, $g_{max}$ and $\Delta_{peak}$, for the C-Dur and C-DC cases. In the C-Dur case (*Equation (7)*), as $P$ increased, $\varphi_{LP}$ decayed and approached 0 like $1/P$. In contrast, in the C-DC case (*Equation (8)*), $\varphi_{LP}$ approached its lower limit $\Delta_{peak} \cdot DC$, as $P$ increased, and thus behaved very differently than in the C-Dur case.

We used these equations to describe $g_{max}$ as a function of $P$ (for any given $\Delta_{peak}$) so that LP maintained a constant phase $\varphi_c$, (*Equation (10)* for the C-DC case). Alternatively, $\Delta_{peak}$ could be given as a function of $P$ (for any given $g_{max}$, *Equation (11)* for the C-DC case). We used these derivations to compare how phase constancy depends on $g_{max}$ or $\Delta_{peak}$ in the C-DC case. A comparison of these two cases can be seen in *Figure 9A*, where either $g_{max}$ (green) or $\Delta_{peak}$ (blue) is varied to keep $\varphi_{LP}$ constant at $\varphi_c$=0.34 across different $P$ values. (The red curve is the depressing case, described below.) As the figure shows, phase constancy can be achieved by varying either parameter, but each parameter produces a different range of $P$ across which phase is maintained.

These equations and their corresponding counterparts for the C-Dur case can be used to calculate the range of $P$ values over which changing $\Delta_{peak}$ (from 0 to 1) can maintain a constant phase $\varphi_c$. If $\Delta P$ denotes the range of $P$ values for which phase can be constant, it is straightforward to show that $\Delta P_{DC} > \Delta P_{Dur}$ (compare blue and black curves in *Figure 9B and C*; see Materials and methods for derivation).

Two additional points are notable in *Figure 9C*. First, the lower bound on $\varphi_{LP}$ for which phase constancy can occur is smaller in the C-Dur (black) than the C-DC (blue) case. This is because we have assumed that in the C-DC case the LP neuron cannot fire during inhibition and therefore the constant value of DC produces a lower limit for $\varphi_{LP}$. Second, for $\varphi_c$ larger than ~0.5, $\Delta P$ is larger for the C-Dur case. This occurs because *Equation (12)* can no longer be satisfied when $\varphi_c$ is large. That is, with constant duty cycle, it is not possible to produce an arbitrarily large follower neuron phase, but with constant duration, any large phase is attainable if the cycle period is not much larger than the synaptic duration. These findings are consistent with our experimental results described above (see *Figure 8*).

The pacemaker synaptic input to the LP neuron shows short-term synaptic depression (*Rabbah and Nadim, 2007*). In a previous modeling study, we explored how the phase of a follower neuron was affected when the inhibitory synapse from an oscillatory neuron to this follower had short-term synaptic depression (*Manor et al., 2003*). In that study the role of the parameter $\Delta_{peak}$ was not considered. We now consider how the presence of short-term synaptic depression influences phase constancy by changing both $g_{max}$ and $\Delta_{peak}$. As stated in the Materials and methods (*Equation (16)*), the effect of synaptic depression on synaptic strength can be obtained as $g_{max} = \bar{g}_{max} \cdot s_{max}(P)$, where $s_{max}$ is an increasing function whose value approaches one as $P$ increases. This indicates that the synapse becomes stronger due to more recovery from depression at longer cycle periods. When synaptic depression dictates how $g_{max}$ varies with $P$ and $\Delta_{peak}$ also varies with $P$ and $g_{max}$ (*Equation (11)*), the simultaneous changes in $g_{max}$ and $\Delta_{peak}$ (red) greatly increase the range of $P$ values over which $\varphi_{LP}$ is constant (*Figure 9A*).

Note that the C-DC case with short-term depression spans a larger range of $P$ values than the non-depressing case (*Figure 9B*). Similarly, the range of $P$ values for which phase can be maintained is larger than the non-depressing case across $\varphi_{LP}$ values, except where $\varphi_{LP}$ is so large that the depressing synapse operates outside its dynamic range (*Figure 9C*). These results are consistent with our experimental results, indicating that although phase constancy can be achieved when either $g_{max}$ or $\Delta_{peak}$ increases with $P$, a concomitant increase of both - which could occur for example with

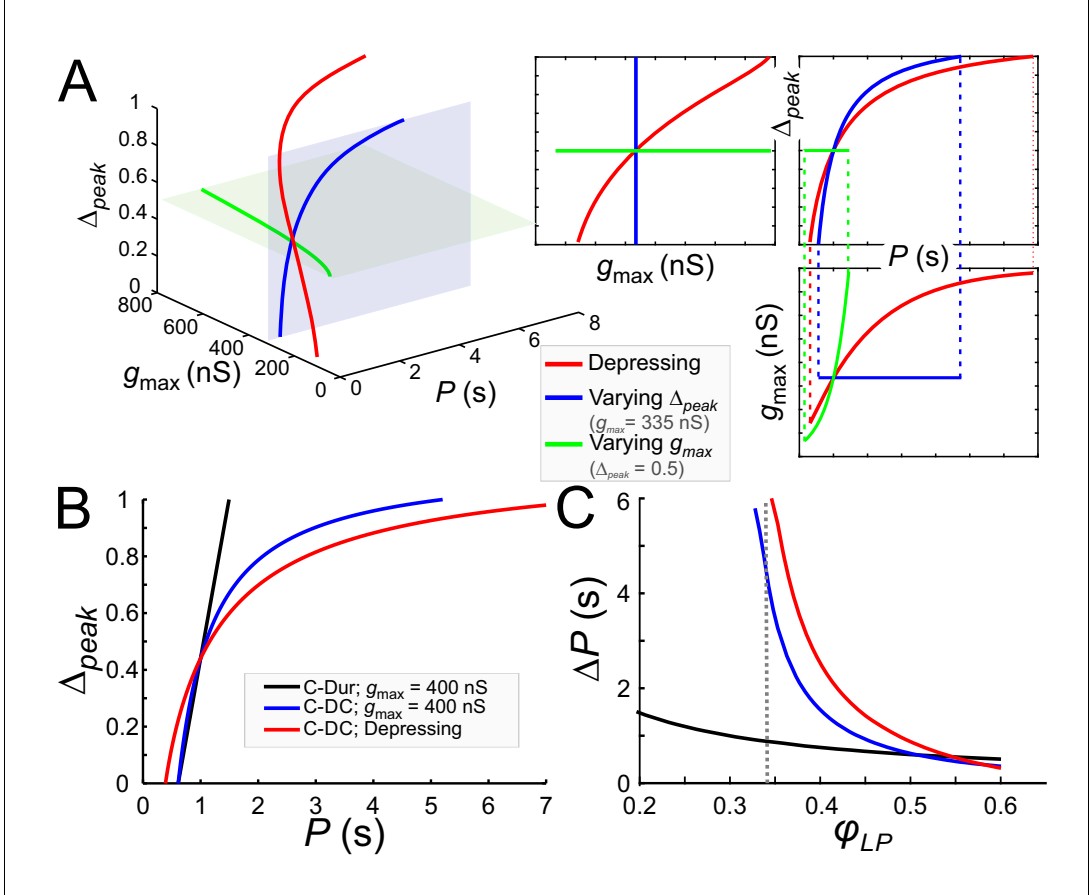

**Figure 9.** Model prediction of the range of phase constancy. (A) For the C-DC case, a constant phase of $\varphi_{LP}$=0.34 can be maintained across a range of cycle periods $P$ when $g_{max}$ is constant (at 335 nS; blue plane) and $\Delta_{peak}$ varies from 0 to 1 according to **Equation (11)** (blue), or when $\Delta_{peak}$ is fixed (at 0.5; green plane) and $g_{max}$ varies from 200 to 800 nS according to **Equation (10)**. Alternatively, $g_{max}$ and $\Delta_{peak}$ can covary to maintain phase, as in a depressing synapse, where $g_{max}$ varies with $P$ according to **Equation (16)**, and $\Delta_{peak}$ is calculated for each $P$ and $g_{max}$ value according to **Equation (11)**. As seen in the 2D coordinate-plane projections of the 3D graph (right three graphs), the range of $P$ values for which phase constancy is achieved is largest when $g_{max}$ and $\Delta_{peak}$ covary (dotted lines show limits of $P$ for phase constancy). The depressing synapse conductance value is chosen to be 335 nS at $P$ = 1 s. (B, C) A comparison between the C-DC and C-Dur cases shows that in the latter case a constant phase of $\varphi_{LP}$ can be maintained across a larger range of $P$ values when $\Delta_{peak}$ increases with $P$ (and $g_{max}$ is fixed at 400 nS) according to **Equation (11)**. The relationship of $\Delta_{peak}$ and $P$ is shown in B for $\varphi_{LP}$=0.34. (C) shows the range of $P$ values ($\Delta P$) of cycle periods for which phase remains constant at any value of $\varphi_{LP}$. If $g_{max}$ also varies with $P$, as in a depressing synapse (red; **Equation (16)**), the range of $P$ values for which phase is constant is further increased. (Dotted line: $\varphi_{LP}$=0.34.).

DOI: https://doi.org/10.7554/eLife.46911.012

a depressing synapse - greatly expands the range of $P$ values for which a constant phase is maintained.

## Discussion

### The importance of phase in oscillatory networks

A common feature of oscillatory networks is that the activities of different neuron types are restricted to specific phases of the oscillation cycle. For example, different hippocampal and cortical neurons are active in at least three distinct phases of the gamma rhythm (*Hájos et al., 2004*; *Hasenstaub et al., 2005*), and distinct hippocampal neuron types fire at different phases of the theta rhythm and sharp wave-associated ripple episodes (*Somogyi and Klausberger, 2005*).

Experimental studies quantify the latency of neural activity with respect to a reference time in the cycle, but in most cases, these latencies are normalized and reported as phase. Distinct neuron types

can maintain a coherent activity phase, despite wide variations in the network frequency (30–100 Hz for gamma rhythms, 4–7 Hz for theta rhythms, and 120–200 Hz for sharp wave-associated ripple episodes). Phase-specific activity of different neuron types is proposed to be important in rhythm generation (*Wang, 2010*), and indicates the necessity of precise timing for producing proper circuit output and behavior (*Kopell et al., 2011*). For example, phase locking of spike patterns to oscillations is important for auditory processing, single cell and network computations and Hebbian learning rules (*Kayser et al., 2009*; *McLelland and Paulsen, 2009*; *Panzeri et al., 2010*). For brain oscillations, phase relationships may provide clues about the underlying circuit connectivity and dynamics, but a behavioral correlate of varying frequencies is not obvious. In contrast, the activity phase of distinct neuron types in rhythmic motor circuits is a tangible readout of the timing of motor neurons and muscle contractions, thus defining phases of movement (*Grillner and El Manira, 2015*; *Kiehn, 2016*; *Le Gal et al., 2017*; *Bidaye et al., 2018*). Because meaningful behavior depends crucially on proper activity phases, whether neurons maintain their activity phase in face of changes in frequency simply translates to whether the movement pattern changes as it speeds up or slows down.

## Determinants of phase

In oscillatory networks, the activity phases of different neuron types depend to different degrees on the precise timing and strength of their synaptic inputs (*Oren et al., 2006*). Our results from noise current injections showed that the timing of the LP neuron is strongly dependent on the timing of inputs it receives. Dynamic clamp injection of realistic or triangular conductance waveforms with different periods ($P$) indicated that $\varphi_{LP}$ was largely determined by the duration of the synaptic input. $\varphi_{LP}$ changed substantially with $P$ when inputs had constant duration, but much less when inputs had a constant duty cycle, that is when duration scaled with $P$. However, our experiments also showed that inputs of constant duty cycles alone are insufficient for phase constancy. $\varphi_{LP}$ decreased with $P$ even with a constant duty cycle of inputs, but increased with either synaptic strength ($g_{max}$) or peak phase of the synaptic input ($\Delta_{peak}$). The increase in $\varphi_{LP}$ had similar sensitivity to $g_{max}$ and $\Delta_{peak}$, and therefore a larger sensitivity to a simultaneous increase in both. Consequently, it was possible to keep $\varphi_{LP}$ constant over a wide range of cycle periods by increasing both parameters with $P$.

The fact that an increase in $g_{max}$ with $P$ promotes phase constancy is biologically relevant, as short-term depression in pyloric synapses means that synaptic strength indeed increases with $P$ (*Manor et al., 1997*). Previous modeling studies show that short-term synaptic depression of inhibitory synapses promotes phase constancy (*Nadim et al., 2003*; *Bose et al., 2004*), largely because of longer recovery times from depression at larger values of $P$.

The finding that an increase of $\Delta_{peak}$ with $P$ promotes phase maintenance is somewhat surprising, as we have previously shown that $\Delta_{peak}$ in LP actually decreases with $P$ (*Manor et al., 1997*; *Tseng et al., 2014*). On the face of it, this suggests that an increase in $\Delta_{peak}$ is not a strategy employed in the intact circuit. However, the caveat is that such results may critically depend on the cause of the change in $P$, either experimentally or biologically. While in our current study we varied $\Delta_{peak}$ with direct conductance injection into LP, previous results were obtained by changing the waveform and period of the presynaptic pacemaker neurons. When $P$ is changed in an individual preparation by injecting current into or voltage-clamping the pacemakers, phase of follower neurons is not particularly well maintained. An example of this is shown in *Figure 1*, where $\varphi_{LP}$ values fall between constant phase and constant duration and, additionally, all pyloric neurons show behavior that falls between constant phase and constant latencies (*Hooper, 1997b*; *Hooper, 1997a*). This may reflect that neurons are not keeping phase particularly well when the only cause of changing $P$ is the presynaptic input. This is supported by the observation that even during normal ongoing pyloric activity, phases change with cycle-to-cycle variability of $P$ in individual preparations (*Bucher et al., 2005*). However, it does not preclude the possibility that $\Delta_{peak}$ plays an important role in stable phase relationships when $P$ differs because of temperature, neuromodulatory conditions, or inter-individual variability (discussed below).

It is noteworthy that a change in the synaptic strength or peak phase with $P$ is not peculiar to graded synapses. The fact that short-term synaptic plasticity can act as a frequency-dependent gain control mechanism is well known for many spike-mediated synaptic connections. In bursting neurons, the presence of a combination of short-term depression and facilitation in the same spike-mediated

synaptic interaction could also result in changes in the peak phase of the summated synaptic current as a function of burst frequency and duration, and the intra-burst spike rate (*Markram et al., 1998*).

The mathematical model in the current study provides mechanistic explanations for several of our experimental findings. First, it can be used to produce a quantitative measure of phase, given the values of $g_{max}$, $\Delta_{peak}$ and $P$. Thus, these equations can be used to compare the C-DC and C-Dur cases, which match our experimental results. They show that, for most phase values, the C-DC case provides a larger range of cycle periods at which phase constancy can occur. Second, these equations provide the activity phase no matter how the pacemaker synaptic input duration changes with cycle period. For instance, our experiments were conducted by changing synaptic input through sampling individual values of the parameter pairs $g_{max}$ and $\Delta_{peak}$, and then calculating the resulting phase. We then used fitting to find level sets of constant phase (*Figure 8*). In contrast, when we combined our mathematical derivation here with previous results on the role of short-term synaptic depression (*Bose et al., 2004*), we could demonstrate how a neuronal circuit can naturally follow a level set of phase (*Equation (7), (8), (15), (16)*). Moreover, we showed that the combined increase in $g_{max}$ and $\Delta_{peak}$ with $P$ produces a larger range of periods for phase constancy than increasing either parameter alone. In short, this mathematical formulation produces a simple quantitative distillation of our experimental results.

In this study, we did not explore the role of the intrinsic properties of the LP neuron on its phase. In separate experiments, we simultaneously measured post-inhibitory rebound properties in LP neurons and the levels of voltage-gated ionic currents (the transient potassium current $I_A$ and the hyperpolarization-activated inward current $I_h$) that influence rebound spiking. These data were not included in this study for brevity and because they showed that the timing of post-inhibitory spiking was relatively stable across preparations. Therefore, we would expect the contribution of intrinsic properties in controlling the timing of the LP neuron burst onset to be relatively small. However, this result does not generalize to all follower neurons. For example, the follower ventral dilator (VD) and PY neurons have a much higher levels of $I_A$, which in turn has a larger effect on the timing of post-inhibitory spiking. In a set of computational studies, we addressed the role of $I_A$ in determining the burst phase in response to periodic inputs (*Zhang et al., 2008*; *Zhang et al., 2009*) and in conjunction with short-term depression in the synaptic input (*Bose et al., 2004*). An experimental clarification of the relative contribution of intrinsic properties vs. synaptic input could be done with controlled dynamic clamp synaptic input, such as those used in the current study, injected in PY or VD neurons. Such a data set would fittingly complement the results of the current study to elucidate more general rules in determining the activity phase of neurons in an oscillatory network.

## Phase relationships in changing temperatures

An interesting case is provided by the observation that phases are remarkably constant when pyloric rhythm frequency is changed with temperature. *Tang et al. (2012)* report a fourfold decrease in $P$ of the pyloric rhythm between 7 and 23° C. In this study, none of the pyloric phases changed significantly, and it is worth noting that under conditions of changing temperatures, the relationships between $P$, $g_{max}$, and $\Delta_{peak}$ appeared to be fundamentally different from when $P$ is changed at a constant temperature. Presynaptic voltage trajectories scaled with changing $P$, and $\Delta_{peak}$ of postsynaptic currents was independent of $P$, in contrast to the decrease described at constant temperature (*Manor et al., 1997*; *Tseng et al., 2014*). Amplitudes of synaptic potentials did not change with temperature, despite an increase in synaptic current amplitudes with increasing temperature (and associated decrease in $P$). This is in contrast to the positive relationship between $g_{max}$ and $P$ that results from synaptic depression at a constant temperature (*Manor et al., 1997*). Therefore, it appears that the likely substantial effects of temperature on synaptic dynamics and ion channel gating are subject to a set of compensatory adaptations different from when $P$ is changed at constant temperature.

## Variability and slow compensatory regulation of phase

Phase maintenance in the face of changing $P$ in an individual animal requires the appropriate short-term dynamics of synaptic and intrinsic neuronal properties. The fact that characteristic (and therefore similar) phase relationships can also be observed under the same experimental conditions across individual preparations is a different conundrum, particularly when $P$ can vary substantially, as

is true for brain oscillations (*Hájos et al., 2004*; *Hasenstaub et al., 2005*; *Somogyi and Klausberger, 2005*). Phases show different degrees of variability across individuals in a variety of systems, for example leech heartbeat (*Wenning et al., 2018*), larval crawling in *Drosophila* (*Pulver et al., 2015*), and fictive swimming in zebrafish (*Masino and Fetcho, 2005*), but in all these cases phases are not correlated with $P$. In the pyloric rhythm, phases are also variable to a degree across individuals, but not correlated with the mean $P$, which varies >2 fold (*Bucher et al., 2005*; *Goaillard et al., 2009*). This phase constancy occurs despite considerable inter-individual variability in ionic currents, and is considered the ultimate target of slow compensatory regulation, that is homeostatic plasticity (*Marder and Goaillard, 2006*; *Ma and LaMotte, 2007*; *Marder et al., 2014*). Slow compensation can also be observed directly when rhythmic activity is disrupted by decentralization, and subsequently recovers to similar phase relationships over the course of many hours (*Luther et al., 2003*). It is interesting to speculate if our findings about how synaptic parameters must change to keep phase constant would hold across individuals with different mean $P$. The prediction would be coordinated positive correlations of both $g_{max}$ and $\Delta_{peak}$ with $P$.

Synaptic inputs to the LP neuron show considerable variability across preparations (e.g. *Figure 3B*), which mirrors the variability seen in the levels of voltage-gated ionic currents in pyloric neurons (*Schulz et al., 2006*). We did not address the role and extent of variability in this study, because a proper analysis of variability required us to first establish the mechanisms that give rise to a consistent output, in this case phase constancy. Based on our findings regarding the influence of synaptic parameters on phase, a natural next step is to explore whether the variability of different parameters defining the synaptic input influences variability of phase or, alternatively, whether variability in some synaptic parameters may be irrelevant to phase or restrained by the postsynaptic neuron.

## Phase relationships under different neuromodulatory conditions

The flipside of the question how neurons maintain phase is the question how their phase can be changed. In motor systems, in particular, changes in phase relationships are functionally important to produce qualitatively different versions of circuit output, for example to produce different gaits in locomotion (*Vidal-Gadea et al., 2011*; *Grillner and El Manira, 2015*; *Kiehn, 2016*). The activity of neural circuits is flexible, and much of this flexibility is provided by modulatory transmitters and hormones which alter synaptic and intrinsic neuronal properties (*Brezina, 2010*; *Harris-Warrick, 2011*; *Jordan and Sławińska, 2011*; *Bargmann, 2012*; *Marder, 2012*; *Bucher and Marder, 2013*; *Nadim and Bucher, 2014*). The pyloric circuit is sensitive to a plethora of small molecule transmitters and neuropeptides which affect cycle frequency and phase relationships (*Marder and Bucher, 2007*; *Stein, 2009*; *Daur et al., 2016*). Indeed, extensive research has indicated the role of amine modulation of synaptic strength and neuronal firing phase in the pyloric circuit, and how amine modulation of synaptic and intrinsic firing properties changes firing phases (*Johnson et al., 2003*; *Gruhn et al., 2005*; *Johnson et al., 2005*; *Peck et al., 2006*; *Harris-Warrick and Johnson, 2010*; *Harris-Warrick, 2011*; *Kvarta et al., 2012*). With respect to our findings, any given neuromodulator could act presynaptically to alter $P$, duration, or duty cycle on the one hand, and $g_{max}$ and $\Delta_{peak}$ on the other. In addition, the neuromodulator could affect the postsynaptic neuron's properties and alter its sensitivity to any of these parameters. Therefore, our findings could not just further our understanding of how phase can be maintained across different rhythm frequencies, but also provide a framework for testing if and how changes in synaptic dynamics may contribute to altering phase relationships under different neuromodulatory conditions.

## Materials and methods

Adult male crabs (*Cancer borealis*) were acquired from local distributors and maintained in aquaria filled with chilled (10–13°C) artificial sea water until use. Crabs were prepared for dissection by placing them on ice for 30 min. The dissection was performed using standard protocols as previously described (*Tohidi and Nadim, 2009*; *Tseng and Nadim, 2010*). The STNS, including the four ganglia (esophageal ganglion, two commissural ganglia, and the STG) and their connecting nerves, and the motor nerves arising from the STG, were dissected from the stomach and pinned into a Sylgard (Dow-Corning) lined Petri dish filled with chilled saline. The STG was desheathed, exposing the somata of the neurons for intracellular impalement. Preparations were superfused with chilled (10-

13°C) physiological *Cancer* saline containing: 11 mM KCl, 440 mM NaCl, 13 mM $CaCl_2 \cdot 2H_2O$, 26 mM $MgCl_2 \cdot 6H_2O$, 11.2 mM Trizma base, 5.1 mM maleic acid with a pH of 7.4.

Extracellular recordings were obtained from identified motor nerves using stainless steel electrodes, amplified using a differential AC amplifier (A-M Systems, model 1700). One lead was placed inside a petroleum jelly well created to electrically isolate a small section of the nerve, the other right outside of it. For intracellular recordings, glass microelectrodes were prepared using the Flaming-Brown micropipette puller (P97; Sutter Instruments) and filled with 0.6 M $K_2SO_4$ and 20 mM KCl. Microelectrodes used for membrane potential recordings had resistances of 25–30 MΩ; those used for current injections had resistances of 15–22 MΩ. Intracellular recordings were performed using Axoclamp 2B and 900A amplifiers (Molecular Devices). Data were acquired using pClamp 10 software (Molecular Devices) and the Netsuite software (Gotham Scientific), sampled at 4–5 kHz and saved on a PC using a Digidata 1332A (Molecular Devices) or a PCI-6070-E data acquisition board (National Instruments).

Individual pyloric neurons were impaled and identified via their membrane potential waveforms, correspondence of spike patterns with extracellular nerve recordings, and interactions with other neurons within the network (*Weimann et al., 1991*).

## Constructing realistic graded IPSC waveforms

Inhibitory postsynaptic currents (IPSCs) were recorded from LP neurons during the ongoing rhythm using two-electrode voltage clamp and holding the LP neuron at −50 mV, far from the IPSC reversal potential of ∼ −80 mV (*Figure 3A*). We refer to the total current measured in the voltage-clamped LP neuron during the activity of the PD and PY neurons as a synaptic current for the following reasons: 1, the after blocking the PTX-sensitive component of the pacemaker synapses, the LP neuron produces tonic spiking activity (see, for example *Figure 2B*), and 2, holding the LP neuron at different voltages (e.g. −60 or −110 mV) produces a similarly shaped current, but with a different amplitude or reversed sign (at −110 mV).

When the LP soma is voltage clamped at −50 mV, the axon (which is electrotonically distant from the soma) produced action potentials following the synaptic inhibition from the PY neuron and the pacemaker neurons. The onset of the LP neuron action potentials (recorded in the current trace) was used to calculate the mean IPSC for each experiment averaging the IPSCs over 10–20 cycles. The IPSC waveforms were then extracted by normalizing both the amplitude and the duration of the mean IPSC.

## Driving the LP neuron with noise current

In these experiments, the preparation was superfused in *Cancer* saline plus $10^{-5}$ M picrotoxin (PTX; Sigma Aldrich) for 30 min to block the synaptic currents to the LP neuron. The removal of synaptic inhibition onto LP neurons changed the activity of these neurons from bursting to tonic firing. Then, noise current, generated by the Ornstein-Uhlenbeck (O-U) process (*Lindner, 2019*), 60 min using the Scope software (available at http://stg.rutgers.edu/Resources.html, developed in the Nadim laboratory). The baseline of the noise current was adjusted by adding DC current so that it can provide enough inhibition to produce silent periods alternating with bursts of action potentials. The O-U process was defined as

$$dX_t = -\frac{1}{\tau} X_t dt + \sigma dW_t.$$

The parameters used for noise injection were τ = 10 to 20 ms, σ = 200 pA and a DC current of −200 to −100 pA. In these experiments, we defined bursts as groups of at least two action potentials with inter-spike intervals < 300 ms, following a gap of at least 300 ms.

## Driving the LP neuron with realistic or triangular IPSC waveforms in dynamic clamp

The dynamic clamp current was injected using the Netclamp software (Netsuite, Gotham Scientific). We pharmacologically blocked synaptic inputs from the pacemaker AB and follower PY neurons to the LP neuron by superfusing the perparation in *Cancer* saline plus $10^{-5}$ M picrotoxin (PTX; Sigma Aldrich) for 30 min. This treatment does not block the cholinergic synaptic input from the PD

neurons for which no clean pharmacological blocker is known. Although the PD neuron input has some influence on the LP neuron activity, this input only constitutes <20% of the total pacemaker synapse and cannot drive oscillations in the follower LP neuron.

The LP neuron was driven in PTX with an artificial synaptic current in dynamic clamp. The synaptic current was given as

$$I_{syn} = g_{syn}\left(V_{LP} - E_{syn}\right)$$

where the synaptic conductance $g_{syn}$ was a pre-determined waveform, repeated periodically with period $P$, and $E_{syn}$ was the synaptic reversal potential set to $-80$ mV (*Zhao et al., 2011*).

Two sets of dynamic clamp experiments were performed on different animals. In one set of experiments, $g_{syn}$ was set to be a triangular waveform. We measured the effects of four different parameters in these triangle conductance injections (*Figure 1*): peak phase ($\Delta_{peak}$), duration ($T_{act}$), period ($P$ = time between onsets of dynamic clamp synaptic injections), and maximal conductance ($g_{max}$, the peak value of $g_{syn}$). This allowed us to explore which combinations of the different parameters influences the LP phase. Five values for $P$ were used: 500, 750, 1000, 1500, and 2000 ms, which cover the typical range of pyloric cycle periods. Three values of $g_{max}$ were used: 0.1, 0.2 and 0.4 µS, consistent with previous measurements of synaptic conductance (*Zhao et al., 2011*; *Tseng et al., 2014*). The value of $\Delta_{peak}$ was varied to be 0, 0.25, 0.5, 0.75 or 1. In the same experiment, all runs were done in two conditions: with $T_{act}$ constant across different $P$ values (C-Dur case with $T_{act}$ = 300 ms) or with $T_{act}$ changing proportionally to $P$ (C-DC case with duty cycle $DC = T_{act}/P$=0.3).

In the other set of experiments, $g_{syn}$ was a realistic IPSC waveform, based on a pre-recorded IPSC in the LP neuron. In these experiments, $P$ was varied to be 500, 750, 1000, 1250, 1500, or 2000 ms by scaling the realistic waveform in the time direction. In these experiments, $g_{max}$ was set to be 0.1, 0.2, 0.4, 0.6, or 0.8 µS. The LP neuron burst onset delay ($\Delta t$) was measured relative to the onset of the pacemaker component of the synaptic input (identified by the kink in the synaptic conductance waveform) in each cycle. The burst phase was calculated as $\varphi_{LP} = \Delta t/P$. Phase constancy means that $\Delta t$ changed proportionally to $P$. To measure the LP neuron phase with respect to a new reference point, the end of the pacemaker input. This reference point was defined by drawing a horizontal line from the kink on the synaptic waveform that identified the onset of the pacemaker input, and chosing the first intersection point.

## Determining relationship between cycle period (P), synaptic strength (g_max) and LP phase ($\varphi_{LP}$) using the realistic IPSC waveform

We determined how well the mathematical model derived for constant input duty cycles (see *Equation (8)* below), matched the experimental data obtained with realistic IPSC waveforms. To this end, we fit the model to $\varphi_{LP}$ values measured for all values of $g_{max}$ and $P$, using the standard fitting routine 'fit' in MATLAB (Mathworks).

## Sensitivity of $\varphi_{LP}$ to g_max and $\Delta_{peak}$ across all P values

To explore how $g_{max}$ and $\Delta_{peak}$ may interact to influence $\varphi_{LP}$, we examined the sensitivity of $\varphi_{LP}$ to these two parameters, individually and in combination, for all values of $P$ in our data. For each $P$, we computed the mean value of $\varphi_{LP}$ across all experiments, and all values of $g_{max}$ (0.1, 0.2, 0.3 and 0.4 µS) and $\Delta_{peak}$ (0, 0.25, 0.5, 0.75 or 1). (The $\varphi_{LP}$ value for $g_{max}$ = 0.3 µS was obtained in this case by linearly interpolating the values for 0.2 and 0.4 µS.) This produced a 4 by 5 matrix of all values. For each data point in the matrix, we moved along eight directions ($+g_{max}$, $+\Delta_{peak}$, $-g_{max}$, $-\Delta_{peak}$,$+g_{max}$ and $+\Delta_{peak}$, $-g_{max}$ and $-\Delta_{peak}$,$+g_{max}$ and $-\Delta_{peak}$,$+g_{max}$ and $-\Delta_{peak}$). Here "+" denotes increasing and "- "denotes decreasing. We then calculated the change in $\varphi_{LP}$ per unit $g_{max}$ (normalized by 0.4 µS), $\Delta_{peak}$, or both. For example, the sensitivity of $\varphi_{LP}$ when $\Delta_{peak}$ was changed from 0.25 to 0.5 was measured as

$$\frac{\varphi_{LP}\left(\text{at}\,\Delta_{peak} = 0.5\right) - \varphi_{LP}\left(\text{at}\,\Delta_{peak} = 0.25\right)}{0.5 - 0.25}$$

Similarly, the sensitivity of $\varphi_{LP}$ when $g_{max}$ was changed from 0.2 to 0.4 was measured as

$$\frac{\varphi_{LP}(\text{at } g_{max} = 0.4) - \varphi_{LP}(\text{at } g_{max} = 0.2)}{(0.4 - 0.2)/0.4}$$

These data are provided in *Figure 7—source data 1*. As the next step, we averaged the sensitivity along each aligned direction: [$+g_{max}$ and $-g_{max}$]; [$+\Delta_{peak}$ and $-\Delta_{peak}$]; [$+g_{max}$ & $+\Delta_{peak}$ and $-g_{max}$ & $-\Delta_{peak}$]; [$+g_{max}$ & $-\Delta_{peak}$ and $+g_{max}$ & $-\Delta_{peak}$]. This produced the four cardinal directions, shown in *Figure 7*. Finally, we averaged the sensitivity across all $P$ values.

## A model of synaptic dynamics

In the derivation of the model, the firing time of the LP neuron was assumed to be completely determined by its synaptic input. This synaptic conductance ($g_{syn}$) was assumed to rise and fall with distinct time constants. The following holds over one cycle period and therefore time is reset with period $P$ ($t$ (mod $P$)):

$$\frac{dg_{syn}}{dt} = \begin{cases} (g_{max} - g_{syn})\tau_r & t \ (mod P) < t_{peak} \\ -g_{syn}/\tau_s & t \ (mod P) \geq t_{peak} \end{cases} \tag{1}$$

where the time $t_{peak}$, corresponding to $\Delta_{peak}$, is $t_{peak} = \Delta_{peak} T_{act}$. We assumed that LP neuron remained inactive when $g_{syn}$ was above a fixed threshold ($g^*$) less than $g_{max}$. Because the synaptic input is periodic with period $P$, we solved for the minimum and maximum values of $g_{syn}$ in each cycle. The minimum ($g_{lo}$) occurred just before the onset ($t = 0$) of AB/PD activity, whereas the maximum occurred at the peak synaptic phase $\Delta_{peak}$ for the C-Dur case. In the C-DC case, $T_{act} = DC \cdot P$, where $DC$ is the duty cycle (fixed at 0.3 in our experiments).

To calculate $g^*$, we set the value $t = 0$ so that $g_{syn}(0) = g_{lo}$ (and, by periodicity, $g_{syn}(P)=g_{lo}$), and solved the first part of *Equation (1)* where $g_{syn}$ increases until $t = t_{peak}$. This yielded

$$g_{peak} = g_{syn}(t_{peak}) = g_{max} + (g_{lo} - g_{max})e^{-t_{peak}/\tau_r} \tag{2}$$

We then used the second part of *Equation (1)* to track the decay of $g_{syn}$ for $t_{peak} < t < P$:

$$g_{syn}(t) = g_{peak}e^{-(t - t_{peak})/\tau_s} \tag{3}$$

Using *Equation (3)*, we calculated the time $\Delta t$ at which the synaptic conductance $g_{syn}(\Delta t)=g^*$ as follows:

$$g* = g_{peak}e^{-(\Delta t - t_{peak})/\tau_s} \tag{4}$$

Solving *Equation (4)* for $\Delta t$ yielded

$$\Delta t = \tau_s ln\frac{g(t_{peak})}{g*} + t_{peak}.$$

Dividing this equation by $P$ yielded $\varphi_{LP}$:

$$\varphi_{LP} = F(P, g_{max}, \Delta_{peak}) = \frac{\tau_s}{P} ln\frac{g_{peak}}{g*} + \frac{t_{peak}}{P}, \tag{5}$$

where $g_{peak}$ is given by *Equation (2)*. This expression provides a description of the dependence of $\varphi_{LP}$ as a function of $P$, $g_{max}$ and $\Delta_{peak}$. To explore the role of the parameters in this relationship, we made a simplifying assumption that the synaptic conductance $g_{syn}(t)$ rapidly reached its peak (i.e., $\tau_r$ was small), stayed at this value and started to decay at $t = t_{peak}$. In this case $g(t)=g_{max}$ on the interval $(0,t_{peak})$ and the value of $g_{lo}$ is irrelevant. With this assumption, *Equation (5)* reduced to

$$\varphi_{LP} = \frac{\tau_s}{P} ln\frac{g_{max}}{g*} + \frac{t_{peak}}{P}. \tag{6}$$

Substituting $t_{peak} = \Delta_{peak} \cdot T_{act}$ in *Equation (6)*, gave

$$\varphi_{LP} = F(P, g_{max}, \Delta_{peak}) = \frac{1}{P}\left(\tau_s ln\frac{g_{max}}{g*} + \Delta_{peak}T_{act}\right), \tag{7}$$

which we used to describe the LP phase in the C-Dur case. To describe the C-DC case, after substituting $t_{peak} = \Delta_{peak} \cdot DC \cdot P$, we obtained

$$\varphi_{LP} = F\left(P, g_{max}, \Delta_{peak}\right) = \frac{1}{P}\left(\tau_s ln\frac{g_{max}}{g*}\right) + \Delta_{peak}DC. \tag{8}$$

Note that these equations also describe the relationship between $\varphi_{LP}$ with $T_{act}$ in the C-Dur case, and $DC$ in the C-DC case).

*Equations (7), (8)* and can be used to approximate a range of parameters over which $\varphi_{LP}$ is maintained at a constant value $\varphi_c$. To do so, we assumed a specific parameter set, say $\left(\hat{P}, \hat{g}_{max}, \hat{\Delta}_{peak}\right)$, satisfies

$$F\left(\hat{P}, \hat{g}_{max}, \hat{\Delta}_{peak}\right) = \varphi_c,$$

for some fixed phase value, $\varphi_c$. We could now ask whether there are nearby parameters for which phase remains constant, that is $F$ remains equal to $\varphi_c$. The Implicit Function Theorem (*Krantz and Parks, 2012*) guarantees that this is the case, provided certain derivatives evaluated at $\left(\hat{P}, \hat{g}_{max}, \hat{\Delta}_{peak}\right)$ are non-zero, which turns out to be true over a large range of parameters. Since the partial derivative with respect to $\Delta_{peak}$ of $F(P, g_{max}, \Delta_{peak})$ at this point is a non-zero constant equal to $T_{act}/P$ (or $DC$) in the C-Dur (or C-DC) case, there is a function $\Delta_{peak} = h(P, g_{max})$ such that

$$F(P, g_{max}, h(P, g_{max})) = \varphi_c \tag{9}$$

for values of $P$ and $g_{max}$ near $\left(\hat{P}, \hat{g}_{max}\right)$. In other words, the Implicit Function Theorem guarantees that small changes in $P$ and $g_{max}$ can be compensated for by an appropriate choice of $\Delta_{peak}$ in order to maintain a constant LP phase. A similar analysis can be done by solving for $g_{max}$ in terms of $P$ and $\Delta_{peak}$ or by solving for $P$ in terms of $g_{max}$ and $\Delta_{peak}$.

Keeping $g_{max}$ (respectively, $\Delta_{peak}$) constant in these equations allows us to obtain a relationship between $P$ and $\Delta_{peak}$ (respectively, $g_{max}$), for which $\varphi_{LP}$ is kept constant at $\varphi_c$. Consider *Equations (7), (8)* and for fixed values of both $\varphi_{LP}$ (= $\varphi_c$) and $g_{max}$. Then these equations reduce to simple functional relationships where $\Delta_{peak}$ can be expressed as a function of $P$. In the C-DC case, for example, evaluating $\Delta_{peak}$ from *Equation (8)* produces

$$g_{max} = g* \cdot exp\left(\frac{P}{\tau_s}\left(\varphi_c - \Delta_{peak}DC\right)\right) \tag{10}$$

*Equation (10)* describes how $g_{max}$ must vary with $P$ for the system to maintain a constant phase $\varphi_c$ for any given $\Delta_{peak}$.

Alternatively, $\Delta_{peak}$ can be expressed as a function of $P$. In the C-DC case, evaluating $\Delta_{peak}$ from *Equation (8)* produces

$$\Delta_{peak} = \frac{\varphi_c}{DC} - \frac{\tau_s}{DC \cdot P}ln\frac{g_{max}}{g*}, \tag{11}$$

*Equation (11)* can be used to calculate the range of $P$ values over which changing $\Delta_{peak}$ (from 0 to 1) can maintain a constant phase $\varphi_c$. Solving $0 < \Delta_{peak} < 1$ using *Equation (11)* yields

$$\frac{\tau_s}{\varphi_c}ln\frac{g_{max}}{g*} < P_{DC} < \frac{\tau_s}{\varphi_c - DC}ln\frac{g_{max}}{g*} \tag{12}$$

Performing the same procedure in the C-Dur case, we find

$$\frac{\tau_s}{\varphi_c}ln\frac{g_{max}}{g*} < P_{Dur} < \frac{T_{act}}{\varphi_c} + \frac{\tau_s}{\varphi_c}ln\frac{g_{max}}{g*}. \tag{13}$$

The lower limits of the two cases ($P_{DC}$ and $P_{Dur}$) are the same. The upper limit for $P_{DC}$ is larger than that of $P_{Dur}$ if

$$\varphi_c < DC\left(1 + \frac{\tau_s}{T_{act}} ln\frac{g_{max}}{g*}\right). \tag{14}$$

If $\Delta P$ denotes the range of $P$ values that respectively satisfy *Equation (12) or (13)*, then $\Delta P_{DC} > \Delta P_{Dur}$ if the inequality given by holds, which it does for true for $\tau_s$ and $g_{max}$ large enough.

## Adding synaptic depression to the model of synaptic dynamics

In a previous modeling study, we explored how the phase of a follower neuron was affected when the inhibitory synapse from an oscillatory neuron to this follower had short-term synaptic depression (*Manor et al., 2003*). In that study the role of the parameter $\Delta_{peak}$ was not considered. It is straightforward to add synaptic depression to *Equations (7), (8)* and therefore examine how phase is affected if $\Delta_{peak}$ increases with $P$ and synaptic strength also changes with P according to the rules of synaptic depression. We will restrict this section to the C-DC case. A similar derivation can be made for the C-Dur case.

An *ad hoc* model of synaptic depression can be made using a single variable $s_d$ which will be a periodic function that denotes the extent of depression and takes on values between 0 and 1 (*Bose et al., 2004*). $s_d$ decays during the AB/PD burst (from time 0 to $T_{act}$, indicating depression) and then recovers during the inter-burst interval (from $T_{act}$ to $P$, indicating recovery). Thus, $s_d$ can be described by an equation of the form:

$$\frac{ds_d}{dt} = \begin{cases} -s_d/\tau_\beta & t\,(mod\,P) \leq T_{act} \\ (1-s_d)/\tau_\alpha & T_{act} < t\,(mod\,P) < P \end{cases}$$

Using periodicity, it is straightforward to show that the maximum value of $s_d$, which occurs at the start of the AB/PD burst, is given by:

$$s_{max}(P) = \frac{1 - e^{-P(1-DC)/\tau_\alpha}}{1 - e^{-P(1-DC)/\tau_\alpha} e^{-DC \cdot P/\tau_\beta}}. \tag{15}$$

Note that $s_{max}$ is a monotonically increasing function with values between 0 and 1. Its value approaches one as $P$ increases, indicating that the synapse becomes stronger. For a complete derivation and description, see *Bose et al. (2004)*. The effect of synaptic depression on synaptic strength can be obtained by setting

$$g_{max} = \bar{g}_{max} \cdot s_{max}(P) \tag{16}$$

where $s_{max}$ is given by *Equation (15)*.

## Software, analysis and statistics

Data were analyzed using MATLAB scripts to calculate the time of burst onset and the phase. Statistical analysis was performed using Sigmaplot 12.0 (Systat). Significance was evaluated with an $\alpha$ value of 0.05, error bars and error values reported denote standard error of the mean (SEM) unless otherwise noted.

## Acknowledgements

We thank Drs. Horacio Rotstein and Eric Fortune for helping with the initial MATLAB scripts in the analysis. This study was supported by NIH MH060605 and NSF DMS1122291.

## Additional information

### Funding

| Funder | Grant reference number | Author |
| --- | --- | --- |
| National Institutes of Health | MH060605 | Dirk M Bucher<br>Farzan Nadim |
| National Science Foundation | DMS1122291 | Amitabha Bose |

The funders had no role in study design, data collection and interpretation, or the decision to submit the work for publication.

## Author contributions
Diana Martinez, Amitabha Bose, Data curation, Formal analysis, Validation, Investigation, Methodology, Writing—original draft; Haroon Anwar, Data curation, Formal analysis, Methodology, Writing—review and editing; Dirk M Bucher, Resources, Writing—review and editing; Farzan Nadim, Conceptualization, Resources, Software, Formal analysis, Supervision, Funding acquisition, Validation, Investigation, Visualization, Methodology, Writing—original draft, Project administration, Writing—review and editing

## Author ORCIDs
Diana Martinez ![ORCID] https://orcid.org/0000-0003-0982-4092
Haroon Anwar ![ORCID] https://orcid.org/0000-0002-3079-4812
Farzan Nadim ![ORCID] https://orcid.org/0000-0003-4144-9042

## Decision letter and Author response
Decision letter https://doi.org/10.7554/eLife.46911.014
Author response https://doi.org/10.7554/eLife.46911.015

# Additional files

## Data availability
Source data files have been provided for Figures 2 and 7.

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
