## [Decision Letter]

Thank you for submitting your article "Short-term synaptic dynamics control the activity phase of neurons in an oscillatory network" for consideration by *eLife*. Your article has been reviewed by Ronald Calabrese as the Senior Editor and Reviewing Editor, and three reviewers. The following individual involved in review of your submission has agreed to reveal their identity: Astrid A Prinz (Reviewer #1).

The reviewers have discussed the reviews with one another and the Reviewing Editor has drafted this decision to help you prepare a revised submission.

Summary:

This manuscript addresses the role of synaptic input dynamics in controlling the phase of activity of neurons in the oscillatory pyloric motor network of the crustacean stomatogastric nervous system. Through systematic experimental and mathematical, analysis, the paper explores and explains the influence of various synaptic timing and amplitude parameters on activity phasing and phase maintenance, using the LP follower neuron. A major conclusion of the work is that intrinsic cellular response properties are not sufficient to ensure proper cell activity phasing and instead need to be complemented by appropriate synaptic input timing and dynamics to achieve a functional activity pattern. This and other findings should generalize well to other oscillatory neuronal networks.

Essential revisions:

There are a few concerns that must be addressed, however, before acceptance. The expert reviews are provided, but as a guide to orchestrating a revision, we provide the following points.

1) The mathematical model should be better integrated into the overall paper and discussion should include what we learn from the model. One reviewer (#2) was unable to see the value of the model given how peripheral it seemed to the results/discussion and so suggested a network model. Such a model would be beyond the scope of the present paper, but this review underscores how important it is for the authors to show the relevance of the mathematical model to the results.

2) Reviewer #1 was concerned about the treatment of variability in the paper. This concern can be addressed by analyzing, describing, displaying, and especially discussing cycle-to-cycle and animal-to-animal information that should already be present in the data underlying this paper.

3) Reviewer #2 was concerned that there is no mention of whether/how intrinsic properties of the LP neuron might contribute to its phasing.

4) Reviewer #3 was concerned that the authors make clear the predominantly graded nature of synaptic transmission in the pyloric network, and that the authors discuss the work of the Harris-Warrick lab that addresses amine modulation of synaptic strength and neuronal firing phase in the pyloric network, and how amine modulation of synaptic and intrinsic firing properties changes firing phases.

[Editors’ note: The separate reviews follow for reference; for the purposes of publication, the minor comments sent to the authors have not been included.]

*Reviewer #1:*

This manuscript addresses the role of synaptic input dynamics onto neurons in oscillatory circuits in controlling the phase of activity of those neurons in the ongoing rhythmic circuit activity. Through an experimental, mathematical, and analysis tour de force, the paper comprehensively explores and explains the influence of various synaptic timing and amplitude parameters on activity phasing and phase maintenance, using the LP follower neuron in the crustacean pyloric circuit as a testbed. A major conclusion of the work is that intrinsic cellular response properties are not sufficient to ensure proper cell activity phasing and instead need to be complemented by appropriate synaptic input timing and dynamics to achieve functionally meaningful oscillations. This and other findings should generalize well to other oscillatory neuronal systems.

While some individual findings presented here confirm prior results by the authors and others, the manuscript presents the most comprehensive framework, to date, that organizes a multitude of findings and parameter dependencies of neuronal oscillatory activity into a coherent picture supported by the fruitful combination of experimentation and mathematical analysis. It will likely become a go-to publication in the area of cellular and circuit oscillation analysis.

My only major comment concerns the treatment of variability in the paper, and can likely be addressed by analyzing, describing, displaying, and especially discussing cycle-to-cycle and animal-to-animal information that should already be present in the data underlying this paper, rather than requiring additional experiments. I note that with the exception of data points in Figure 2H-K and error bars (are these SD or SE?) in Figure 6 and Figure 7, all individual data presented is already in one or several ways normalized or scaled. It is therefore almost impossible for the reader to get a sense how variable rhythmic activity really is between individuals. To derive insights about general operational principles of the pyloric circuit (and oscillatory circuits in general), it makes total sense and provides the clearest answers to average data across animals, as is done in this manuscript. However, as the authors correctly state, proper phasing of neuronal activity in an oscillatory circuit can be of vital importance for physiologically functional performance of a motor system. The level at which achieving functional phase relationships matters most is the individual level, not the population average – in other words, for a crab to properly process its food it matters that its own pyloric circuit produces the right phasing, not whether crab pyloric circuits on average do so. Furthermore, work from the Calabrese lab shows that synaptic properties in an oscillatory circuit can be tuned to postsynaptic neuron properties on an individual basis to ensure that every animal implements a good solution for the problem of achieving proper phasing. All I am suggesting is that the authors provide some more detail about inter-individual variability in their text and figures (perhaps by showing some individual raw traces at extreme ends of the spectrum), and, more importantly, that they discuss animal-to-animal variability, its biological significance, and the advantages and disadvantages of their approach of processing their results largely in average form.

I further note that cycle-to-cycle variability is not mentioned, described, quantified, displayed, or discussed in the paper at all. However, in the context of this work I find a discussion of variability at the animal-to-animal level more pertinent than at the cycle-to-cycle level.

*Reviewer #2:*

This manuscript addresses the question of how phase is maintained within a rhythmically active circuit. The simplest model that could be used to address this question is a two-cell network consisting of a pacemaker and a follower neuron. This is well embodied by the PD-LP synapse in the stomatogastric system. The authors use a combination of electrophysiology and the dynamic-clamp to determine how peak synaptic conductance and the phase of the synaptic input combine to maintain phase across periods. The use of the dynamic-clamp allows for a substantive exploration of the parameter space composed of Period, peak synaptic conductance and peak phase interact. The data are compelling and suggest that the combined effect of g_max and peak phase are important for phase constancy even though individually they don't appear to be. I think the sensitivity analysis in their Figure 7 is insightful because it is likely that these types of combinations of parameters are what will be important for phase constancy in other systems in which exhaustive measure of intrinsic properties will be prohibitive. For example, a lot of work has been done to try and "unwrap" the individual components of rhythmic inhibition in the hindlimb locomotor network. A similar analysis in that system could inform the relative timing of excitation and inhibition necessary for rhythmic output.

In terms of substantive concerns, my biggest concern is the use of the mathematical model. As far as an explanatory tool ("to get a better understanding of our experimental results"), it is helpful, but in my opinion, it doesn't add to the overall outcome. Perhaps a rewrite of that section, or a simulation of a network of the PD-LP synapse could add to the analysis. Furthermore, the model results are not discussed in any great detail in the discussion, which detracts from their use in the Results section.

The second substantive concern regards the use and numbering of the equations in the Results section and Materials and methods section for the model. They aren't numbered properly, and they are introduced in the Results section sort of mid-stream from the methods derivation. Perhaps leaving it out of the Results section or referring to just the equations would work better. In alignment with the above concern, this may allow for a more focused explanation of how the parameters in the model explain the experimental results.

The final substantive concern is that there is no mention of intrinsic properties of the LP neuron. Although not asking for a measure of them or an inclusion of new experiments, a statement about how the follower neuron's intrinsic properties might contribute to phase would provide a final context for the current results. Presumably the activity of the muscles innervated by LP also need to be properly phased relative to those innervated by PD, and as such the LP neuron may make a contribution to phase via its integration of the synaptic input. As noted, that remains an untested hypothesis, but it is known that intrinsic properties of hindlimb motor neurons contribute to their output.

*Reviewer #3:*

This an important and interesting paper that carefully examines the parameters of synaptic transmission that contribute to the phase maintenance of neuronal firing across different neural network periods. The authors use experimental and computational methods to determine the contribution of specific parameters of synaptic transmission to firing phase constancy of neurons in a model central pattern generator network. They systematically manipulate experimentally and in a mathematical model the duration, timing and amplitude of synaptic currents to determine how each parameter works separately and together to maintain firing phase. These results should be of interest to neuroscientists studying small and large ensembles of neural networks that oscillate. This a complicated paper but the results and figures are mostly well explained.

In the Introduction introduce the concept that the neurons studied use graded as well as action potential evoked synaptic inhibition. That will clarify the results.

In the last paragraph of the Discussion section, the importance of change in neuronal phase relationships for proper network function is raised, The paper could note the work of the Harris-Warrick lab that addresses amine modulation of synaptic strength and neuronal firing phase in the pyloric network, and how amine modulation of synaptic and intrinsic firing properties changes firing phases.

---

## [Author Response]

Essential revisions:There are a few concerns that must be addressed, however, before acceptance. The expert reviews are provided, but as a guide to orchestrating a revision, we provide the following points.1) The mathematical model should be better integrated into the overall paper and discussion should include what we learn from the model. One reviewer (#2) was unable to see the value of the model given how peripheral it seemed to the results/discussion and so suggested a network model. Such a model would be beyond the scope of the present paper, but this review underscores how important it is for the authors to show the relevance of the mathematical model to the results.

The mathematical model sections are now completely revised in the Results section and Materials and methods section and a new section has been added to the Discussion section to address the findings of the model.

2) Reviewer #1 was concerned about the treatment of variability in the paper. This concern can be addressed by analyzing, describing, displaying, and especially discussing cycle-to-cycle and animal-to-animal information that should already be present in the data underlying this paper.

This is a major issue for us too and the subject of a manuscript in preparation. Please see the response to the comment of the reviewer.

3) Reviewer #2 was concerned that there is no mention of whether/how intrinsic properties of the LP neuron might contribute to its phasing.

Please see the response to the comment of the reviewer.

4) Reviewer #3 was concerned that the authors make clear the predominantly graded nature of synaptic transmission in the pyloric network, and that the authors discuss the work of the Harris-Warrick lab that addresses amine modulation of synaptic strength and neuronal firing phase in the pyloric network, and how amine modulation of synaptic and intrinsic firing properties changes firing phases.

Done and done.

Reviewer #1:[…] My only major comment concerns the treatment of variability in the paper, and can likely be addressed by analyzing, describing, displaying, and especially discussing cycle-to-cycle and animal-to-animal information that should already be present in the data underlying this paper, rather than requiring additional experiments. I note that with the exception of data points in Figure 2H-K and error bars (are these SD or SE?) in Figure 6 and Figure 7, all individual data presented is already in one or several ways normalized or scaled. It is therefore almost impossible for the reader to get a sense how variable rhythmic activity really is between individuals. To derive insights about general operational principles of the pyloric circuit (and oscillatory circuits in general), it makes total sense and provides the clearest answers to average data across animals, as is done in this manuscript. However, as the authors correctly state, proper phasing of neuronal activity in an oscillatory circuit can be of vital importance for physiologically functional performance of a motor system. The level at which achieving functional phase relationships matters most is the individual level, not the population average – in other words, for a crab to properly process its food it matters that its own pyloric circuit produces the right phasing, not whether crab pyloric circuits on average do so. Furthermore, work from the Calabrese lab shows that synaptic properties in an oscillatory circuit can be tuned to postsynaptic neuron properties on an individual basis to ensure that every animal implements a good solution for the problem of achieving proper phasing. All I am suggesting is that the authors provide some more detail about inter-individual variability in their text and figures (perhaps by showing some individual raw traces at extreme ends of the spectrum), and, more importantly, that they discuss animal-to-animal variability, its biological significance, and the advantages and disadvantages of their approach of processing their results largely in average form.I further note that cycle-to-cycle variability is not mentioned, described, quantified, displayed, or discussed in the paper at all. However, in the context of this work I find a discussion of variability at the animal-to-animal level more pertinent than at the cycle-to-cycle level.

We thank the reviewer for these comments. We realized that addressing (animal-to-animal) variability within the context of phase constancy requires us to first deal with the synaptic mechanisms that influence phase. Dr. Anwar is lead author on a follow-up study that focuses only on the issue of variability. In that study, we examine how variability of synaptic parameters reflects on the variability of the LP neuron phase. We are in complete agreement with the findings of the Calabrese lab on this topic but beg the patience of the reviewer on a proper treatment of variability in this context in a follow up study.

We now address the issue of variability in the Discussion section.

Reviewer #2:[…] In terms of substantive concerns, my biggest concern is the use of the mathematical model. As far as an explanatory tool ("to get a better understanding of our experimental results"), it is helpful, but in my opinion, it doesn't add to the overall outcome. Perhaps a rewrite of that section, or a simulation of a network of the PD-LP synapse could add to the analysis. Furthermore, the model results are not discussed in any great detail in the Discussion section, which detracts from their use in the Results section.The second substantive concern regards the use and numbering of the equations in the Results section and Materials and method sections for the model. They aren't numbered properly, and they are introduced in the Results section sort of mid-stream from the methods derivation. Perhaps leaving it out of the Results section or referring to just the equations would work better. In alignment with the above concern, this may allow for a more focused explanation of how the parameters in the model explain the experimental results.

The modeling sections of the Results section and Materials and methods section are completely redone and, as the reviewer suggested, all equations were moved to the Materials and methods section.

The final substantive concern is that there is no mention of intrinsic properties of the LP neuron. Although not asking for a measure of them or an inclusion of new experiments, a statement about how the follower neuron's intrinsic properties might contribute to phase would provide a final context for the current results. Presumably the activity of the muscles innervated by LP also need to be properly phased relative to those innervated by PD, and as such the LP neuron may make a contribution to phase via its integration of the synaptic input. As noted, that remains an untested hypothesis, but it is known that intrinsic properties of hindlimb motor neurons contribute to their output.

The reviewer is correct, and this is a topic of a separate study by led Dr. Anwar. We now address this point in the Discussion section.

Reviewer #3:[…] In the Introduction introduce the concept that the neurons studied use graded as well as action potential evoked synaptic inhibition. That will clarify the results.

Done.

In the last paragraph of the Discussion section, the importance of change in neuronal phase relationships for proper network function is raised, The paper could note the work of the Harris-Warrick lab that addresses amine modulation of synaptic strength and neuronal firing phase in the pyloric network, and how amine modulation of synaptic and intrinsic firing properties changes firing phases.

Added to the last paragraph of the Discussion section.